# FFB: A Fair Fairness Benchmark for In-Processing Group Fairness Methods

**Xiaotian Han[1]\*  Jianfeng Chi[2]  Yu Chen[3]  Qifan Wang[2]  Han Zhao[4]  Na Zou[5]  Xia Hu[6]**
[1]Texas A&M University    [2]Meta AI    [3]Anytime AI    [4]UIUC    [5]University of Houston
[6]Rice University
han@tamu.edu    {jianfengchi, wqfcr}@meta.com    ychen@anytime-ai.com
hanzhao@illinois.edu    nzou2@uh.edu    xia.hu@rice.edu

## ABSTRACT

This paper introduces the Fair Fairness Benchmark (FFB), a benchmarking framework for in-processing group fairness methods. Ensuring fairness in machine learning is important for ethical compliance. However, there exist challenges in comparing and developing fairness methods due to inconsistencies in experimental settings, lack of accessible algorithmic implementations, and limited extensibility of current fairness packages and tools. To address these issues, we introduce an open-source standardized benchmark for evaluating in-processing group fairness methods and provide a comprehensive analysis of state-of-the-art methods to ensure different notions of group fairness. This work offers the following key contributions: the provision of flexible, extensible, minimalistic, and research-oriented open-source code; the establishment of unified fairness method benchmarking pipelines; and extensive benchmarking, which yields key insights from $45,079$ experiments, $14,428$ GPU hours. We believe that our work will significantly facilitate the growth and development of the fairness research community. The benchmark is available at https://github.com/ahxt/fair_fairness_benchmark.

## 1 INTRODUCTION

Machine learning models trained on biased data have been found to perpetuate and even exacerbate the bias against historically underrepresented and disadvantaged demographic groups when deployed (Mehrabi et al., 2021; Pessach & Shmueli, 2022; Caton & Haas, 2020; Wan et al., 2023). As a result, concerns about fairness have gained significant attention, especially as applications of these models expand to high-stakes domains such as criminal justice, hiring process, and credit scoring. To mitigate such algorithmic bias, a variety of fairness criteria and algorithms have been proposed, which impose statistical constraints on the model to ensure equitable treatment under the respective fairness notions (Hsu et al., 2022; Chai et al., 2022; Reddy et al., 2021b). However, a fair and objective comparison between the proposed and existing algorithms to enforce fairness can be difficult due to the following reasons:

- *Hard to compare the performance of two objectives: utility*[1] and fairness. Often, there is a trade-off between these two objectives (McNamara et al., 2019; Zhao & Gordon, 2022; Xian et al., 2023). Besides, the instability of the fairness performance of those methods during the training process can complicate the pursuit of an optimal balance (Xian et al., 2023).

- *Inconsistent experimental settings (Madras et al., 2018; Louizos et al., 2016)*: Fairness methods can also be hindered by variations in dataset preprocessing and the use of different backbones. These discrepancies can lead to inconsistent (Table 1) and unfair comparison, further complicating the comparison of methods.

- *Stable and customizable implementation of commonly used fairness methods are not accessible.* The fairness methods are often implemented in different programming languages and frameworks, complicating the reproduction and the comparative analysis of various fairness approaches.

---

\*This work was done while the first author was an intern at Meta.

[1]We use utility to represent the performance of the downstream tasks, which is commonly used in fairness community (Dwork et al., 2012; Li & Liu, 2022; Baumann et al., 2022).

- *Current fairness packages (Bellamy et al., 2018; Bird et al., 2020) and tools often suffer from a lack of extensibility.* This can make it difficult for researchers and practitioners to build upon existing methods and develop new ones.

This work aims to facilitate the growth and development of the fairness research community by addressing existing challenges, promoting more accessible and reproducible methods for fairness implementation, and establishing efficient benchmarking techniques. To achieve these goals, we develop a standardized benchmark for evaluating and comparing in-process group fairness methods, which we make open-source. We also conduct comprehensive experiments and analysis of fairness methods. The **major contributions** of our benchmark are summarized as follows:

- **Extensible, Minimalistic, and Research-oriented Open-Source Code**: We offer open-source implementation for all preprocessing, methods, metrics, and training results, thus facilitating other researchers to utilize and build upon this work. The Fair Fairness Benchmark is publicly available, making it easy for researchers and practitioners to use and contribute to.

- **Unified Fairness Method Benchmarking Pipelines**: Our benchmark includes a unified fairness method development and evaluation pipeline, with three key components: First, we provide a thorough statistical and experimental analysis of widely-used fairness datasets and identity some widely-used datasets unsuitable for studying fairness issues; second, we standardize preprocessing for these datasets, ensuring consistent, comparable evaluations; lastly, we present a range of bias mitigation algorithms and comprehensive evaluation metrics for group fairness.

- **Comprehensive Benchmarking and Detailed Obsevations**: We conduct comprehensive experiments on 14 datasets (each with two sensitive attributes), 6 utility metrics and 9 fairness metrics. We run a total of 45,079 experiments. Our experiments yield the following key insights: ① Not all widely used fairness datasets stably exhibit fairness issues. *(Important for evaluation)*. ② The HSIC achieves the best utility-fairness trade-off among the tested methods. *(Not a popular baseline before)* ③ Utility-fairness trade-offs are generally controllable by the hyperparameter. *(Despite adversarial debiasing method)* ④ Stopping training while learning rate is lower enough is effective. *(Practical strategy)*.

**Scope of this Work.** We focus on the problem of in-processing group fairness, which is defined as discrimination against demographic groups. We consider the fairness methods in the context of binary classification and binary sensitive attributes.

**Notation.** The dataset is denoted as $\{(\mathbf{x}_i, s_i, y_i)_{i=1}^N\}$, where $N$ is the number of samples. For the sample $i$ in the dataset, $\mathbf{x}_i \in \mathbb{R}^d$ is non-sensitive attributes, $s_i \in \{0, 1\}$ represents the binary sensitive attribute, and $y_i \in \{0, 1\}$ is the label of the downstream task. We use $\hat{y} \in \{0, 1\}$ to denote the predicted label of the downstream task, which is obtained by thresholding the output of a machine learning model $f(\mathbf{x}) : \mathbb{R}^d \to [0, 1]$ with trainable parameter $\theta$. Accordingly, we use $X, Y, S$ and $\hat{Y}$ to denote the random variables.

## 2 Why is this Benchmark Needed?

Ensuring fairness in algorithmic predictions is crucial in the field of machine learning. However, fairly comparing the current fairness method is challenging due to inconsistencies in data pre-processing and a lack of flexibility in existing fairness packages. This section aims to analyze the critical issue of current fairness methods and discuss the urgent need for a benchmark to address these challenges.

**Current fairness packages lack flexibility for researchers.** AIF360 (Bellamy et al., 2018) and Fair-Learn (Bird et al., 2020) are two well-known fairness packages that have successfully mitigated bias in machine learning algorithms. As popular open-source Python packages, they provide practitioners with toolkits for detecting and mitigating bias in their models and evaluating model fairness. AIF360 is a comprehensive fairness toolkit offering a variety of algorithms and metrics for addressing bias in machine learning models. FairLearn also provides multiple algorithms and fairness metrics for assessing model performance. While both packages are highly recognized within the fair machine learning community, they may not give researchers the desired flexibility for research purposes. We provide a more in-depth comparison between AIF360, FairLearn, and FFB in Appendix A.

**Inconsistent experimental setting leads to unfair comparison.** Prior research has often experienced inconsistencies in data pre-processing and train test split, which has led to divergent performance

Table 2: Fairness definitions and metrics used in the experiments. Appendix B lists a more comprehensive list of fairness metrics and their details. Their simple code implementations are at this url.

| Abbreviation | Name | Formal Definition |
|---|---|---|
| dp (Dwork et al., 2012) | Demographic/Statistical Parity | $P(\hat{Y} \mid S = 0) = P(\hat{Y} \mid S = 1)$ |
| prule (Zafar et al., 2017) | $p$-Rule | $P(\hat{Y} = 1 \mid S = 1)/P(\hat{Y} = 1 \mid S = 0)| \leq p/100$ |
| ppv (Chouldechova, 2017) | Predictive Parity Value Parity | $P(Y = 1 \mid \hat{Y}, S = 0) = P(Y = 1 \mid \hat{Y}, S = 1)$ |
| bnegc (Kleinberg et al., 2016) | Balance for Negative Class | $\mathbb{E}[f(X) \mid Y = 0, S = 0] = \mathbb{E}[f(X) \mid Y = 0, S = 1]$ |
| bposc (Kleinberg et al., 2016) | Balance for Positive Class | $\mathbb{E}[f(X) \mid Y = 1, S = 0] = \mathbb{E}[f(X) \mid Y = 1, S = 1]$ |
| eopp (Hardt et al., 2016) | Equality of Opportunity | $P(\hat{Y} \mid S = 0, Y = 1) = P(\hat{Y} \mid S = 1, Y = 1)$ |
| eodd (Hardt et al., 2016) | Equalized Odds | $P(\hat{Y} \mid S = 0, Y = y) = P(\hat{Y} \mid S = 1, Y = y), y \in \{0, 1\}$ |
| abcc (Han et al., 2023) | Area Between CDF Curves | $\int_0^1 |F_0(x) - F_1(x)| \, \mathrm{d}x$ |

results that hinder comparison and reproducibility. Minor variations in data preparation and dataset split can significantly impact the performance of machine learning algorithms. Due to these issues, the reported accuracy of tabular data (Adult,German) varies in different papers (Madras et al., 2018; Zemel et al., 2013; Edwards & Storkey, 2015; Feldman et al., 2015; Louizos et al., 2016) shown in Table 1. To tackle these issues, we propose a standardized and consistent data pre-processing and split approach, including data normalization, outlier removal, and the implementation of a uniform train-test split ratio.

**Sufficient and in-depth analysis of fairness methods is urgently needed.** The current fairness method lacks a comprehensive comparison, such as the training curves and stability, the influence of the utility-fairness trade-off parameters, and how to select the best utility-fairness trade-offs during the training process. Our benchmark addresses these shortcomings by offering more in-depth and multifaceted analysis. To present a more thorough understanding of fairness methods, we investigate the training stability, model performance under various fairness constraints, and the selection of best-performing models.

Table 1: The reported accuracy of tabular data varies in different papers.

| Paper | Adult | German |
|---|---|---|
| Madras et al. (2018) | $\approx 0.85$ | — |
| Zemel et al. (2013) | $\approx 0.70$ | $\approx 0.69$ |
| Edwards & Storkey (2015) | $\approx 0.83$ | — |
| Feldman et al. (2015) | $\approx 0.68$ | $\approx 0.69$ |
| Louizos et al. (2016) | $\approx 0.82$ | $\approx 0.78$ |
| $\Delta_{\max}$ | 0.17 | 0.09 |
| $\Delta$Percentage | 20% | 13% |

## 3   FFB: FAIR FAIRNESS BENCHMARK

To overcome the above limitations of the previous methods, we introduce the Fair Fairness Benchmark (FFB), a extensible, minimalistic, and research-oriented fairness benchmark package. Compare to other fairness packages, FFB codebase has the following **main characteristics**:

- **Extensible**: We provide the source code for fairness methods implementation, allowing researchers to modify, extend, and tailor these methods to suit their specific requirements and implement new ideas upon our code. This fosters a more customizable approach to developing fairness methods.

- **Minimalistic**: We focus on delivering core fairness methods and allowing users to understand the fundamental techniques comprehensively without being overwhelmed by unnecessary complexity. This approach ensures that users can easily implement and integrate our methods into their existing workflows while maintaining a solid grasp of the underlying concepts.

- **Research-oriented**: We include benchmark datasets and evaluation metrics that facilitate assessing fairness methods. This simplifies the research process, allowing researchers to quickly compare and analyze the effectiveness of different methods in various scenarios.

### 3.1   GROUP FAIRNESS METRICS

To provide a comprehensive comparison for bias mitigating methods, we consider multiple fairness metrics, including demographic parity, $p$-rule, equality of opportunity, equalized odds, the area between CDF curves, *etc.* Table 2 presents the fairness metrics used in Section 4 and Section 5.

Table 3: The summary of the benchmarking datasets. The #nFeat/#cFeat is the number of the numerical/categorical features and the #allFeat is the total number of the features after our preprocessing. The $y_0 : y_1/s_0 : s_1$ is the ratio of two classes of the target label and the sensitive attributes. More details about datasets are presented in Appendix D. The dataset loading codes are at this url.

| Dataset | Task | SensAttr | #Instances | #nFeat | #cFeat | #allFeat | $y_0 : y_1$ | $s_0 : s_1$ (1st) | $s_0 : s_1$ (2nd) |
|---|---|---|---|---|---|---|---|---|---|
| Adult (Kohavi & Becker, 1996) | income | gender, race | $45,222$ | 7 | 5 | 101 | $1 : 0.33$ | $1 : 2.08$ | $1 : 9.20$ |
| German (Dua & Graff, 2017) | credit | gender, age | $1,000$ | 13 | 6 | 58 | $1 : 2.33$ | $1 : 2.23$ | $1 : 4.26$ |
| KDDCensus (Dua & Graff, 2017) | income | gender, race | $292,550$ | 32 | 8 | 509 | $1 : 14.76$ | $1 : 0.92$ | $1 : 8.14$ |
| COMPAS (Larson et al., 2016) | recidivism | age | $6,172$ | 400 | 5 | 405 | $1 : 0.83$ | $1 : 4.25$ | — |
| Bank (Dua & Graff, 2017) | credit | gender, race | $41,188$ | 10 | 9 | 62 | $1 : 0.13$ | $1 : 37.58$ | $1 : 37.58$ |
| ACS-I (Ding et al., 2021) | income | gender, race | $195,665$ | 8 | 1 | 908 | $1 : 0.70$ | $1 : 0.89$ | $1 : 1.62$ |
| ACS-E (Ding et al., 2021) | employment | gender, race | $378,817$ | 15 | 0 | 187 | $1 : 0.84$ | $1 : 1.03$ | $1 : 1.59$ |
| ACS-P (Ding et al., 2021) | public | gender, race | $138,554$ | 18 | 0 | 1696 | $1 : 0.58$ | $1 : 1.27$ | $1 : 1.31$ |
| ACS-M (Ding et al., 2021) | mobility | gender, race | $80,329$ | 20 | 0 | 2678 | $1 : 3.26$ | $1 : 0.95$ | $1 : 1.32$ |
| ACS-T (Ding et al., 2021) | traveltime | gender, race | $172,508$ | 15 | 0 | 1567 | $1 : 0.94$ | $1 : 0.89$ | $1 : 1.61$ |
| CelebA-A (Liu et al., 2015) | attractive | gender, age | $202,599$ | — | — | $48 \times 48$ | $1 : 0.95$ | $1 : 1.40$ | $1 : 0.29$ |
| CelebA-W (Liu et al., 2015) | wavy hair | gender, age | $202,599$ | — | — | $48 \times 48$ | $1 : 2.13$ | $1 : 1.40$ | $1 : 0.29$ |
| CelebA-S (Liu et al., 2015) | smiling | gender, age | $202,599$ | — | — | $48 \times 48$ | $1 : 1.07$ | $1 : 1.40$ | $1 : 0.29$ |
| UTKFace (Zhang et al., 2017) | age | gender, race | $23,705$ | — | — | $48 \times 48$ | $1 : 1.15$ | $1 : 1.10$ | $1 : 1.35$ |

## 3.2 Benchmarking Datasets

To provide a comprehensive comparison of fairness methods, we adopted multiple commonly-used fairness datasets (Le Quy et al., 2022) for our experiments, including tabular and image datasets. Table 3 summarizes the datasets used. they are publicly available and can be downloaded from the corresponding websites. We also present the number of features and the ratio of the target label and sensitive attributes. For example, the target label ratio of KDDCensus is $1 : 14.76$, which is an extremely unbalanced dataset.

## 3.3 Data Preprocessing

To ensure a fair comparison and maintain the reproducibility of the fairness approach, we adhere to a conventional data preprocessing strategy. We apply standard normalization for numerical features while employing one-hot encoding to process the categorical features. We also split the data into training and test sets with random seeds. We use the training set to train the model and the test set to evaluate the model's performance. We use the same data preprocessing strategy for all the datasets.

## 3.4 Benchmarking Fairness Methods

In this section, we introduce the benchmarking methodology employed in our experiments. The benchmarking methods can be classified into three categories: surrogate loss, independence constraints, and adversarial debiasing techniques. In this paper, we focus on in-processing methods for fairness primarily due to the following: 1) They intervene directly in the learning algorithm to ensure fairness, which provides a more nuanced and effective approach to mitigating bias; 2) The emergence of more in-processing techniques designed in deep neural networks calls for systematic comparison; 3) In-processing methods are susceptible to information leakage since they do not require sensitive attributes during inference. In particular, we consider the following three types of in-processing methods. **Gap Regularization** (Chuang & Mroueh, 2020) simplifies optimization by offering a smooth approximation to real loss functions, which are often non-convex or difficult to optimize directly. This approach includes DiffDP, DiffEodd, and DiffEopp. **Independence** introduces fairness constraint into the optimization process to minimize the impact of protected attributes on model predictions while maintaining performance. This approach includes PRemover (Kamishima et al., 2012) and HSIC (Li et al., 2019). **Adversarial Learning**[2] minimizes the utility loss while preventing the adversary from accurately predicting the protected attributes. This approach includes

---

[2]For adversarial learning methods, we use gradient reversal (Ganin et al., 2016) for better training stability.

Table 4: Bias examination for all datasets. We identify the biased dataset specified with sensitive attributes with the reported utility and fairness performance of ERM. ~~Numbers~~ (e.g., ~~$1.35_{\pm1.17}$~~) mean that the bias is too small, indicating is not suitable for fairness assessment. This only reflects our own recommendation and fairness metrics should be appropriately chosen for different applications. The biased datasets are marked with ✔, while the unbiased datasets are marked with ✗. The ✗ indicates that the bias exists but with a large standard deviation. The results are based on 10 trials with varying data splits and training seeds, to ensure reliable outcomes. The table is generated from **910** runs.

| Dataset | SenAttr | Utility | | | | Fairness | | | | | Recommend? |
|---|---|---|---|---|---|---|---|---|---|---|---|
| | | acc | auc | ap | f1 | dp | abcc | prule | eodd | eopp | |
| ~~Bank~~ | Age | $91.17_{\pm0.57}$ | $94.05_{\pm0.19}$ | $62.41_{\pm1.83}$ | $54.31_{\pm11.14}$ | $10.88_{\pm4.27}$ | $10.64_{\pm1.63}$ | $44.48_{\pm5.95}$ | $10.71_{\pm5.68}$ | $6.16_{\pm4.90}$ | ✗ |
| ~~German~~ | Gender | $75.42_{\pm2.03}$ | $78.55_{\pm1.97}$ | $89.21_{\pm1.06}$ | $83.02_{\pm1.54}$ | $7.36_{\pm4.35}$ | $4.89_{\pm1.77}$ | $90.47_{\pm5.74}$ | $14.45_{\pm11.55}$ | $2.74_{\pm1.76}$ | ✗ |
| | Age | $75.19_{\pm2.16}$ | $77.21_{\pm2.60}$ | $88.28_{\pm1.74}$ | $82.90_{\pm1.62}$ | $12.20_{\pm6.02}$ | $10.01_{\pm1.63}$ | $84.44_{\pm7.17}$ | $17.97_{\pm10.71}$ | $8.26_{\pm6.12}$ | ✗ |
| Adult | Gender | $85.35_{\pm0.34}$ | $91.06_{\pm0.34}$ | $78.50_{\pm0.72}$ | $66.78_{\pm0.75}$ | $16.67_{\pm0.69}$ | $18.36_{\pm0.71}$ | $32.54_{\pm2.62}$ | $14.16_{\pm3.12}$ | $7.93_{\pm2.88}$ | ✔ |
| | Race | $85.21_{\pm0.27}$ | $91.10_{\pm0.16}$ | $78.65_{\pm0.35}$ | $66.85_{\pm0.46}$ | $12.23_{\pm0.72}$ | $12.59_{\pm0.60}$ | $41.54_{\pm2.68}$ | $13.12_{\pm2.65}$ | $8.81_{\pm2.93}$ | ✔ |
| COMPAS | Gender | $67.07_{\pm0.80}$ | $72.56_{\pm0.74}$ | $67.99_{\pm0.93}$ | $59.77_{\pm2.27}$ | $13.43_{\pm2.48}$ | $5.80_{\pm1.12}$ | $65.12_{\pm8.25}$ | $19.67_{\pm6.02}$ | $11.54_{\pm4.73}$ | ✔ |
| | Race | $67.13_{\pm1.06}$ | $72.98_{\pm0.59}$ | $68.24_{\pm0.72}$ | $60.58_{\pm3.06}$ | $16.83_{\pm3.48}$ | $8.15_{\pm1.12}$ | $61.83_{\pm4.56}$ | $29.03_{\pm6.66}$ | $20.05_{\pm3.95}$ | ✔ |
| KDDCensus | Gender | $94.88_{\pm0.48}$ | $94.03_{\pm0.04}$ | $99.55_{\pm0.00}$ | $97.32_{\pm0.24}$ | $3.61_{\pm1.60}$ | $5.20_{\pm0.35}$ | $96.35_{\pm1.62}$ | $14.97_{\pm7.11}$ | ~~$0.77_{\pm0.37}$~~ | ✔ |
| | ~~Race~~ | $94.49_{\pm0.78}$ | $94.40_{\pm0.08}$ | $99.57_{\pm0.01}$ | $97.13_{\pm0.38}$ | ~~$1.35_{\pm1.17}$~~ | $3.31_{\pm0.15}$ | $98.64_{\pm1.18}$ | $6.56_{\pm5.83}$ | ~~$0.28_{\pm0.25}$~~ | ✗ |
| ACS-I | Gender | $82.30_{\pm0.12}$ | $90.28_{\pm0.09}$ | $86.02_{\pm0.15}$ | $77.91_{\pm0.19}$ | $9.10_{\pm0.31}$ | $8.27_{\pm0.24}$ | $79.01_{\pm0.65}$ | $3.38_{\pm0.61}$ | ~~$1.75_{\pm0.41}$~~ | ✔ |
| | Race | $82.40_{\pm0.09}$ | $90.40_{\pm0.09}$ | $86.17_{\pm0.14}$ | $78.11_{\pm0.11}$ | $9.81_{\pm0.39}$ | $7.71_{\pm0.29}$ | $77.24_{\pm0.82}$ | $9.72_{\pm0.73}$ | $6.21_{\pm0.48}$ | ✔ |
| ~~ACS-E~~ | ~~Gender~~ | $81.63_{\pm0.12}$ | $88.95_{\pm0.08}$ | $83.12_{\pm0.12}$ | $81.31_{\pm0.16}$ | ~~$0.60_{\pm0.20}$~~ | ~~$0.56_{\pm0.12}$~~ | ~~$98.87_{\pm0.37}$~~ | $10.77_{\pm0.20}$ | ~~$0.90_{\pm0.18}$~~ | ✗ |
| | ~~Race~~ | $81.99_{\pm0.16}$ | $90.00_{\pm0.13}$ | $85.58_{\pm0.24}$ | $81.38_{\pm0.11}$ | ~~$1.42_{\pm0.35}$~~ | ~~$0.99_{\pm0.11}$~~ | ~~$97.29_{\pm0.62}$~~ | $3.48_{\pm0.82}$ | ~~$2.19_{\pm0.45}$~~ | ✗ |
| ~~ACS-P~~ | ~~Gender~~ | $71.92_{\pm0.18}$ | $75.25_{\pm0.16}$ | $67.23_{\pm0.22}$ | $52.93_{\pm0.71}$ | ~~$2.09_{\pm0.61}$~~ | ~~$2.35_{\pm0.16}$~~ | $91.26_{\pm2.52}$ | ~~$2.30_{\pm1.32}$~~ | ~~$1.52_{\pm1.00}$~~ | ✗ |
| | ~~Race~~ | $71.70_{\pm0.22}$ | $75.00_{\pm0.31}$ | $67.01_{\pm0.28}$ | $52.06_{\pm0.54}$ | ~~$0.48_{\pm0.32}$~~ | ~~$1.98_{\pm0.20}$~~ | ~~$97.87_{\pm1.30}$~~ | $4.63_{\pm0.38}$ | $4.03_{\pm0.72}$ | ✗ |
| ~~ACS-M~~ | ~~Gender~~ | $76.81_{\pm0.32}$ | $72.85_{\pm0.36}$ | $88.40_{\pm0.22}$ | $86.54_{\pm0.31}$ | ~~$0.18_{\pm0.17}$~~ | ~~$0.84_{\pm0.12}$~~ | ~~$99.80_{\pm0.19}$~~ | ~~$0.45_{\pm0.51}$~~ | ~~$0.08_{\pm0.09}$~~ | ✗ |
| | ~~Race~~ | $76.98_{\pm0.65}$ | $73.23_{\pm0.61}$ | $88.53_{\pm0.27}$ | $86.70_{\pm0.04}$ | ~~$0.11_{\pm0.17}$~~ | ~~$1.15_{\pm0.17}$~~ | ~~$99.88_{\pm0.18}$~~ | ~~$0.82_{\pm1.13}$~~ | ~~$0.18_{\pm0.29}$~~ | ✗ |
| ACS-T | Gender | $66.36_{\pm0.22}$ | $73.54_{\pm0.18}$ | $71.59_{\pm0.14}$ | $66.51_{\pm0.31}$ | $8.60_{\pm0.45}$ | $5.02_{\pm0.28}$ | $84.65_{\pm0.74}$ | $12.90_{\pm0.82}$ | $5.72_{\pm0.44}$ | ✔ |
| | Race | $66.45_{\pm0.20}$ | $73.64_{\pm0.17}$ | $71.67_{\pm0.19}$ | $66.26_{\pm0.47}$ | $9.62_{\pm0.67}$ | $6.07_{\pm0.22}$ | $83.09_{\pm0.92}$ | $15.19_{\pm1.24}$ | $6.50_{\pm0.99}$ | ✔ |
| CelebA-A | Gender | $78.19_{\pm0.44}$ | $86.67_{\pm0.53}$ | $86.66_{\pm0.64}$ | $79.17_{\pm0.48}$ | $52.39_{\pm1.27}$ | $37.67_{\pm0.98}$ | $30.42_{\pm1.23}$ | $70.84_{\pm3.15}$ | $35.53_{\pm1.82}$ | ✔ |
| | Age | $78.19_{\pm0.44}$ | $86.67_{\pm0.53}$ | $86.66_{\pm0.64}$ | $79.17_{\pm0.47}$ | $41.90_{\pm1.03}$ | $31.15_{\pm1.17}$ | $33.43_{\pm1.71}$ | $37.42_{\pm2.26}$ | $18.83_{\pm1.94}$ | ✔ |
| CelebA-W | Gender | $82.50_{\pm0.76}$ | $88.38_{\pm0.86}$ | $80.38_{\pm1.57}$ | $70.14_{\pm1.64}$ | $33.92_{\pm1.35}$ | $29.52_{\pm1.12}$ | $16.89_{\pm2.01}$ | $52.71_{\pm4.28}$ | $39.62_{\pm3.49}$ | ✔ |
| | Age | $82.50_{\pm0.76}$ | $88.38_{\pm0.86}$ | $80.38_{\pm1.57}$ | $70.14_{\pm1.64}$ | $10.27_{\pm0.71}$ | $10.61_{\pm0.47}$ | $64.59_{\pm2.35}$ | $10.63_{\pm2.18}$ | $6.48_{\pm1.94}$ | ✔ |
| CelebA-S | Gender | $89.95_{\pm3.40}$ | $96.51_{\pm1.84}$ | $96.67_{\pm1.80}$ | $89.08_{\pm6.79}$ | $14.09_{\pm1.08}$ | $13.02_{\pm1.46}$ | $72.76_{\pm3.44}$ | $6.99_{\pm1.16}$ | $6.51_{\pm1.06}$ | ✔ |
| | Age | $89.95_{\pm3.40}$ | $96.51_{\pm1.84}$ | $96.67_{\pm1.80}$ | $89.08_{\pm6.79}$ | $5.91_{\pm0.75}$ | $5.59_{\pm0.70}$ | $88.21_{\pm2.50}$ | $6.46_{\pm1.06}$ | ~~$0.82_{\pm0.80}$~~ | ✔ |
| UTKFace | Gender | $83.34_{\pm0.71}$ | $91.78_{\pm0.57}$ | $91.36_{\pm0.67}$ | $81.66_{\pm0.85}$ | $25.68_{\pm1.88}$ | $20.57_{\pm1.23}$ | $54.61_{\pm2.54}$ | $28.66_{\pm4.12}$ | $17.16_{\pm2.63}$ | ✔ |
| | Race | $83.34_{\pm0.71}$ | $91.78_{\pm0.57}$ | $91.36_{\pm0.67}$ | $81.66_{\pm0.85}$ | $23.25_{\pm1.71}$ | $18.99_{\pm1.22}$ | $59.67_{\pm2.63}$ | $23.07_{\pm3.64}$ | $16.68_{\pm2.70}$ | ✔ |
| Jigsaw | Gender | $68.10_{\pm0.69}$ | $74.81_{\pm0.82}$ | $74.02_{\pm0.92}$ | $65.82_{\pm2.27}$ | $8.17_{\pm3.35}$ | $4.41_{\pm1.34}$ | $83.11_{\pm5.39}$ | $13.35_{\pm6.49}$ | $4.90_{\pm3.94}$ | ✔ |
| | Race | $68.10_{\pm0.69}$ | $74.81_{\pm0.82}$ | $74.02_{\pm0.92}$ | $65.82_{\pm2.27}$ | $4.56_{\pm2.28}$ | ~~$1.74_{\pm0.52}$~~ | $90.25_{\pm4.85}$ | $7.92_{\pm3.95}$ | $3.13_{\pm2.26}$ | ✔ |

AdvDebias (Zhang et al., 2018; Louppe et al., 2017; Beutel et al., 2017; Edwards & Storkey, 2015; Adel et al., 2019) and LAFTR (Madras et al., 2018). The fairness methods are present as follows:

- **ERM** is a standard machine learning method that minimizes the empirical risk of the training data. It is a common baseline for fairness methods.

- **DiffDP, DiffEopp, DiffEodd** are the gap regularization methods for demographic parity, equalized opportunity, and equalized odds (Chuang & Mroueh, 2020). As these fairness definitions cannot be optimized directly, gap regularization is alternative loss and differentiable and can be optimized using gradient descent.

- **PRemover** (Kamishima et al., 2012) (PrejudiceRemover) minimizes the mutual information between the prediction accuracy and the sensitive attributes.

- **HSIC** (Gretton et al., 2005; Baharlouei et al., 2020; Li et al., 2019) minimizes the Hilbert-Schmidt Independence Criterion between the prediction accuracy and the sensitive attributes.

- **AdvDebias** (Zhang et al., 2018; Louppe et al., 2017; Beutel et al., 2017; Edwards & Storkey, 2015; Adel et al., 2019) (adversarial debiasing) maximize a classifier for prediction ability and simultaneously minimizes an adversary to predict sensitive attributes from the predictions.

- **LAFTR** (Madras et al., 2018) is a fair representation learning method that aims to learn an intermediate representation that minimizes the classification loss, reconstruction error, and the adversary's ability to predict the sensitive attributes from the representation.

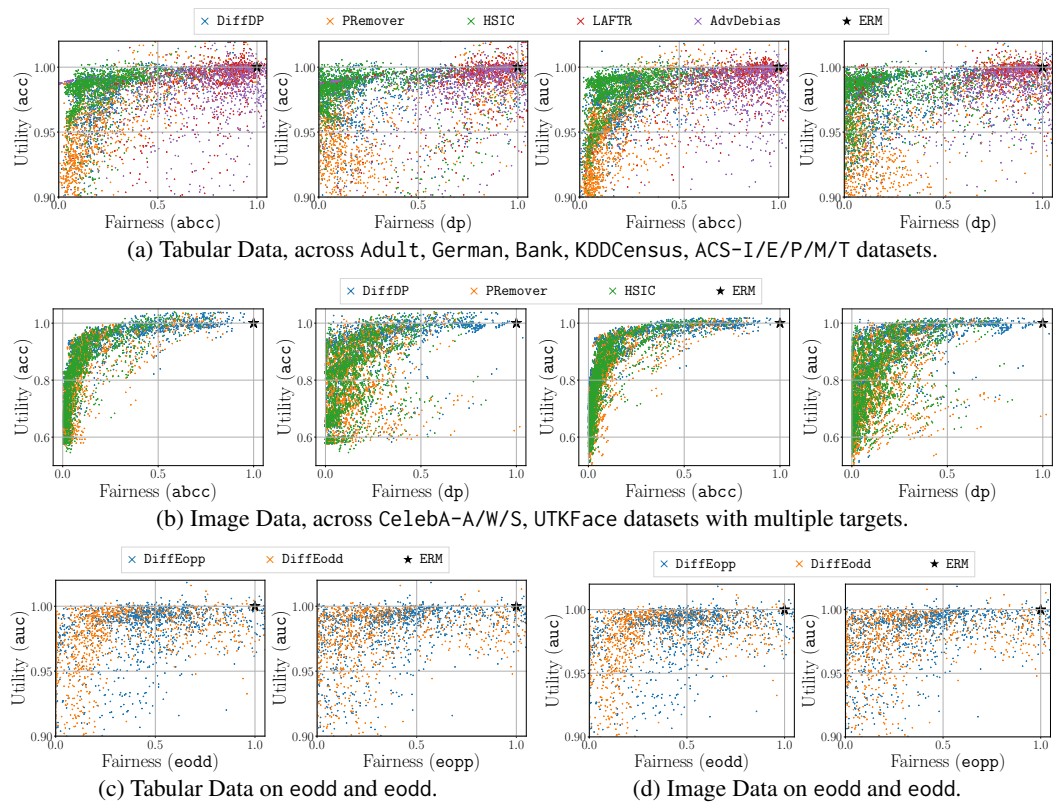

Figure 1: The utility-fairness trade-offs of current fairness methods – DiffDP, PRemover, HSIC, LAFTR, and AdvDebias. To plot the fairness and utility performance in one figure, for each dataset, we normalize the utility (acc,auc) and fairness (abcc, dp) based on the performance of ERM, which is denoted as the point $(1.0, 1.0)$. The figures clearly show that utility-fairness exhibits trader-offs. These figures are generated from a total of **27568** runs.

## 4 BIAS EXAMINATION FOR WIDELY USED FAIRNESS BENCHMARK DATASETS

The presence of bias in the current widely used dataset is not well examined and investigated as it should be, even though such bias can significantly influence the assessment of fairness methods. As such, our work endeavors to delve deeper into this issue by exploring the inherent bias in the widely used dataset. We aim to assess the suitability of these datasets for fairness evaluation critically.

① **Not all widely used fairness datasets stably exhibit fairness issues.** We systematically identify datasets that not only have demonstrated bias but are also prevalently used in fairness research. We found that in some cases, the bias in these datasets is either not consistently present or its manifestation varies significantly. This finding indicates that relying on these datasets for fairness analysis might not always provide stable or reliable results, suggesting the need for more rigorous dataset selection or bias evaluation methodologies in future fairness studies. The biased datasets are marked with ✔ while unbiased ones are with ✗. The ✗ indicates that the bias exists but with a large standard deviation.

## 5 BENCHMARKING CURRENT FAIRNESS METHODS

This section presents comprehensive experiments to benchmark the performance of existing in-processing group fairness methods.

### 5.1 HOW THE BIAS MITIGATING METHODS PERFORM ON UTILITY-FAIRNESS TRADE-OFFS?

In this section, we present the results of experiments conducted to assess the performance of existing in-processing fairness methods in terms of the utility-fairness trade-offs. We analyze how well these methods balance optimizing utility and ensuring fairness in decision-making. To accurately reflect the performance of the different methods, we aggregate the performance across different datasets in

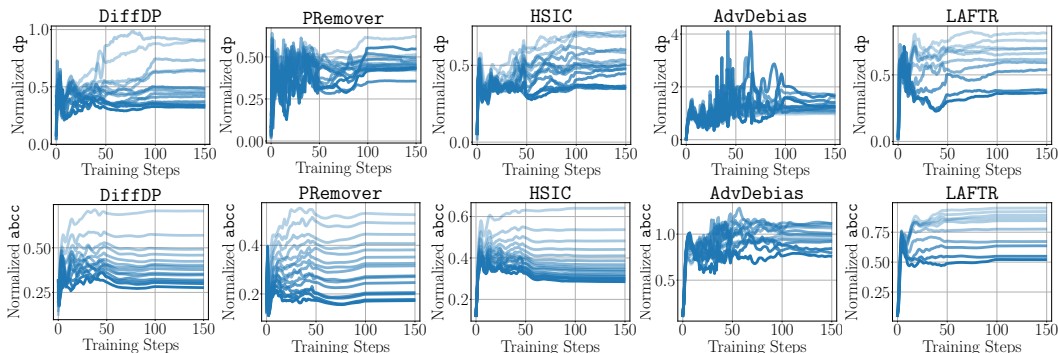

Figure 2: The fairness performance with varying fairness control hyperparameters. The intensity of the color represents the size of the control parameters. In most cases, the **larger** value of control parameters yields better fairness performance, while **small** ones have worse fairness performance. These figures are generated from **13110** runs of experiments.

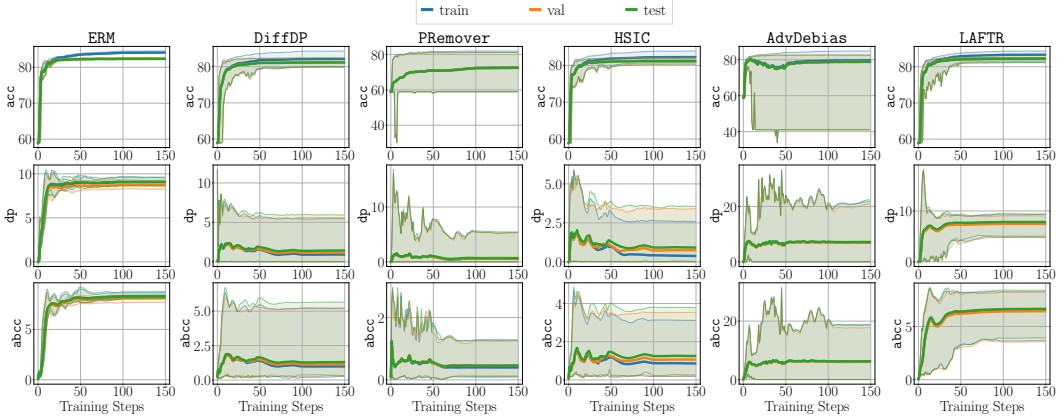

Figure 3: The training curves on tabular dataset. The training curves for fairness metrics typically have larger standard deviation than utility performance, showing the instability of fairness performance.

one figure. To do so, we normalize the utility (acc,auc) and fairness (abcc, dp) performance based on the performance of the ERM. From the results, we make the following major observations:

② **The utility-fairness performance of the current fairness method exhibits trade-offs.** We first present the utility-fairness trade-offs of the existing in-processing fairness methods in Figure 1. We conduct experiments using various in-processing fairness methods and analyze the ability to adjust the trade-offs to cater to specific needs while maintaining a balance between accuracy and fairness.

③ **The HSIC method achieves the best utility-fairness trade-off overall.** The **HSIC method consistently excels in balancing utility and fairness, outperforming other approaches across our tests.** This method, depicted in green in Figure 1, shows particular effectiveness when applied to tabular data. It exhibits a significant ability to improve fairness measures without compromising the precision of utility, maintaining high accuracy in predictions. This quality affirms the robustness of the HSIC method in preserving utility-fairness equilibrium under various conditions. However, when this method is applied to image data, it exhibits a relative performance decline, showing lower fairness and utility scores.

④ **Adversarial debiasing methods exhibit instability.** As illustrated in Figure 1, the utility-fairness points representing the AdvDebias method are scattered randomly across the figures, failing to depict a consistent trade-off pattern. This randomness suggests an inherent instability in adversarial debiasing methods. This inconsistency is further demonstrated in subsequent experiments, where the training curves reveal that these methods are difficult to control effectively.

## 5.2 CAN THE UTILITY-FAIRNESS TRADE-OFFS BE CONTROLLED?

Hereby we investigate the extent to which the utility-fairness trade-offs can be controlled. We conduct experiments using various in-processing fairness methods and analyze the ability of trade-offs.

⑤ **The utility-fairness trade-offs are generally controllable.** The intensity of color in Figure 2 corresponds to the size of the control parameters. With the exception of adversarial debiasing, we find that the performance of most methods can be modulated effectively through the adjustment of hyperparameters. Specifically, larger control hyperparameters tend to yield lower dp and abcc values, indicating enhanced fairness. This implies that the trade-offs between utility and fairness can be actively managed in most cases, providing a crucial degree of flexibility in fairness-oriented data processing tasks. In comparison, the adversarial debiasing method (AdvDebias) is hard to control.

## 5.3 HOW DO UTILITY AND FAIRNESS PERFORMANCE CHANGE DURING TRAINING PROCESS?

In this section, we thoroughly examine the training curves of existing in-processing fairness methods, which are not sufficiently explored in previous studies. We conduct a series of experiments to track the evolution of utility and fairness performance throughout the training process and evaluate the impact of these dynamics on the final results. We presented the training curves in Figures 3 and 4 for tabular data and image data, respectively.

⑥ **The training curves of utility are stable, while those of fairness are unstable.** Results on both tabular and image data show that the standard deviation for fairness metrics is significantly larger than that for utility metrics. Among the fairness methods, LAFTR shows the most stable fairness performance. Even though the value

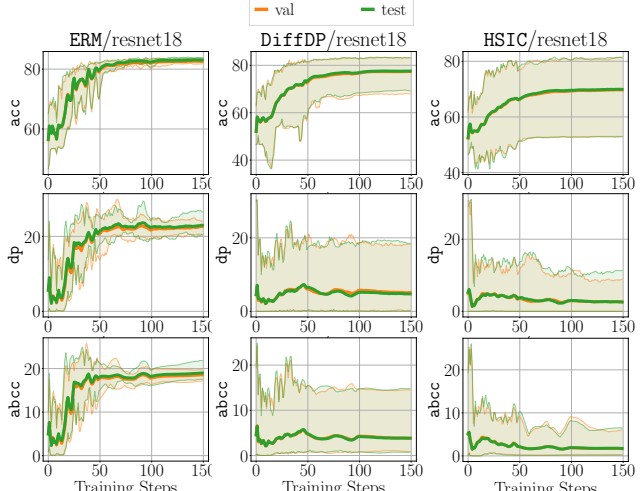

Figure 4: The training curves on image dataset. The results are similar to tabular dataset that training curves for fairness metrics typically have larger standard deviation than utility performance.

of fairness metrics is small, the large standard deviation still suggests unstable fairness performance. The results indicate a future direction focused on enhancing fairness training stability.

⑦ **Stopping training while the learning rate is lower enough proves effective.** In our work, we halt the model training when the learning rate diminishes to a value less than $1e^{-5}$, which is decayed by multiplying by 0.1 every 50 training steps. The utilization of learning rate decay to halt training results in stable fairness metrics, thereby affirming its efficacy and reasonableness.

## 5.4 EXPERIMENT ON TEXT DATA

In this section, we conducted experiments on text data, specifically on comment toxicity classification using the Jigsaw toxic comment dataset (Jigsaw, 2018). The task is to predict whether a comment is toxic or not. Part of this dataset is labeled with identity attributes, such as gender and race. In our study, we regard race and gender and race as sensitive attributes. We follow the setting in (Chuang & Mroueh, 2020) to transform each comment into a vector using BERT (Devlin et al., 2019). Subsequently, we employ an MLP to predict based on the encoded vector. We present the result on the utility-fairness trade-offs in Figure 5, training process in Figure 6, and controllability of trade-offs in Appendix H.

From these results, we observe a similar trend of utility-fairness trade-offs in text data as in tabular and image data in most cases, but there are some slight differences. We made the following observations: 1) Utility and fairness typically exhibit trade-offs. 2) The training stability has an opposite trend compared to tabular/image data. The training curves for utility are unstable, while those for fairness are relatively unstable for most bias mitigation methods.

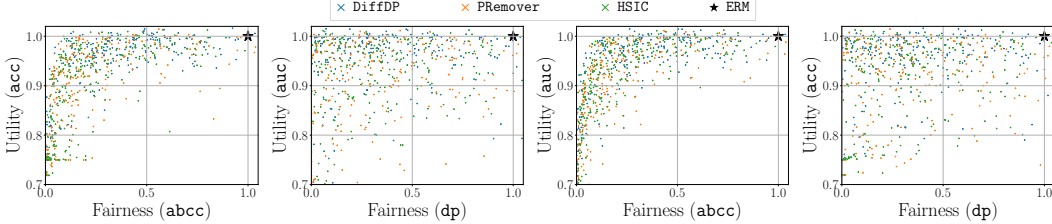

Figure 5: The utility-fairness trade-offs of current fairness methods – DiffDP, PRemover, and HSIC on text data. To plot the fairness and utility performance in one figure, for each dataset, we normalize the utility (acc,auc) and fairness (abcc, dp) based on the performance of ERM, which is denoted as the point $(1.0, 1.0)$. The figures show that utility-fairness exhibits trader-offs.

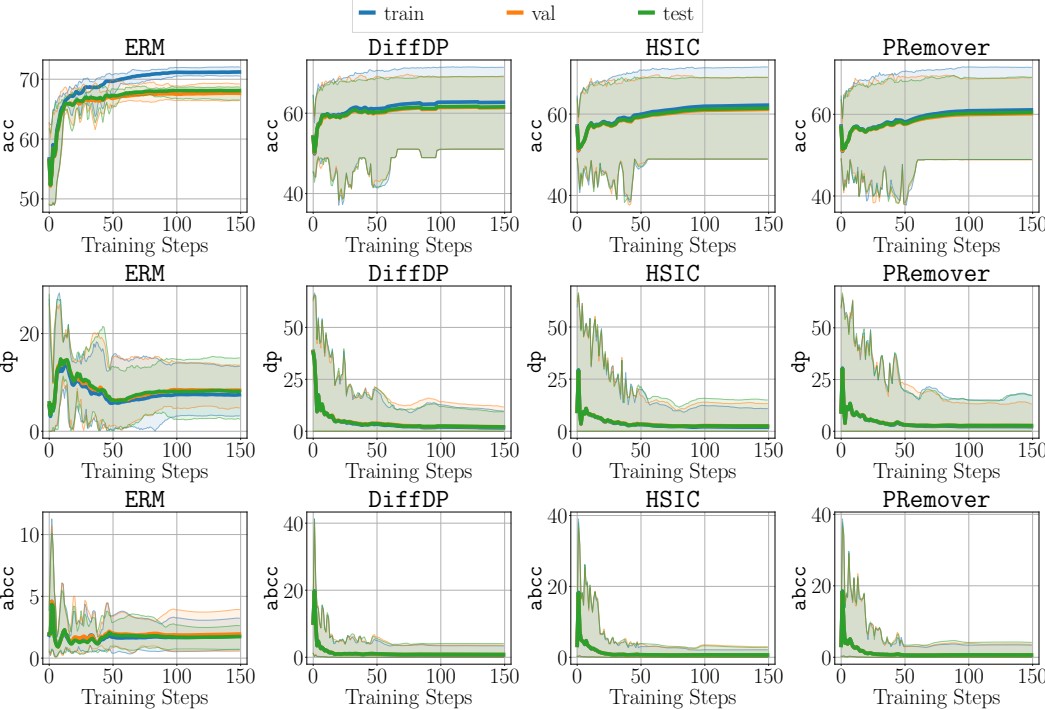

Figure 6: The training curves on text dataset. The training curves for fairness metrics typically have a larger standard deviation than utility performance, showing the instability of fairness performance.

## 6 DISCUSSIONS

This paper introduces Fair Fairness Benchmark (FFB) to benchmark the in-processing group fairness models, offering extensible, minimalistic, and research-oriented open-source code, as well as comprehensive experiments.

**Social Impact.** Our benchmark, with its extensible, minimalistic, and research-oriented open-source code, is designed to facilitate researchers and practitioners to explore and implement fairness methods. Standardized dataset preprocessing and reference baseline implementation will help reduce inconsistencies and make fairness more accessible, especially for beginners in the field. Ultimately, our work aims to stimulate future research on fairness and foster the development of fairness models.

**Limitations.** When the ground truth is biased, relying solely on traditional accuracy can be problematic, as mentioned in (Wick et al., 2019; Zajko, 2022; Ruggieri et al., 2023). We employ multiple metrics (accuracy, average precision, AUC) for evaluating "utility" instead of just relying on accuracy, aiming to alleviate this problem to some extent.

**Future work.** Our plan for subsequent phases of this work involves extending the scope of the FFB to include a wider range of in-processing group fairness methods. Moreover, we intend to incorporate additional definitions of fairness into our evaluations. Exploring methods to measure unbiased accuracy is a promising direction for future research in fairness.

ACKNOWLEDGEMENT

We thank the anonymous reviewers for their constructive suggestions and fruitful discussion. Han Zhao is supported in part by a research grant from the IBM-IL Discovery Accelerator Institute (IIDAI). This work is also supported in part by NSF IIS 1900990, 1939716, 2239257, and 1750074.

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

# Appendix

## Table of Contents

# A    RELATED WORK

**Algorithmic Fairness** Fairness in machine learning has garnered considerable attention in recent years. The goal of fairness in machine learning is to ensure that the machine learning models are fair and unbiased towards an individual or group. Thus, fairness in machine learning can be divided into t categories: group fairness (Dwork et al., 2012; Hardt et al., 2016; Corbett-Davies et al., 2017; Zafar et al., 2017; Madras et al., 2018) and individual fairness (Dwork et al., 2012; Sharifi-Malvajerdi et al., 2019). Group fairness aims to ensure that the machine learning models are fair to different groups of people, while individual fairness aims to "similar individuals should be treated similarly." To mitigate fairness and bias problems in machine learning models, bias mitigation methods can be divided into three categories: pre-processing (Kamiran & Calders, 2012; Calmon et al., 2017), in-processing (Kamishima et al., 2012; Zhang et al., 2018; Madras et al., 2018; Zhang et al., 2022; Buyl & De Bie, 2022; Alghamdi et al., 2022; Shui et al., 2022; Mehrotra & Vishnoi, 2022), and post-processing (Hardt et al., 2016; Jiang et al., 2020). Given that the group fairness metrics are widely adopted in real-world applications and the emergence of more in-processing techniques designed in deep neural network models, we focus on benchmarking in-processing methods for group fairness for neural network models for tabular and image data.

**Fairness Packages and Benchmarks** There are many fairness packages in the literature. Among them, AIF360 (Bellamy et al., 2018) and FairLearn (Bird et al., 2020) are the two most widely used Python packages that provide a set of metrics and algorithms to measure and mitigate bias in machine learning models. They provide a set of metrics to measure the bias of machine learning models, including disparate impact, statistical parity difference, and equal opportunity difference, and a set of algorithms to mitigate the bias of machine learning models. However, both AIF360 and FairLearn implement the bias mitigation algorithms using Scikit-learn (Pedregosa et al., 2011) API design (e.g., the use of `fit()` function) with complicated class inheritance, making the understanding and direct modification of implementation difficult. In comparison, our benchmark decouples the implementation of different bias mitigation algorithms using Pytorch-style (Paszke et al., 2019) training scripts and provides a unified fairness evaluation interface for a comprehensive list of group fairness metrics. One recently proposed benchmark (Reddy et al., 2021a) also aims to benchmark bias mitigation algorithms. However, their benchmark only includes adversarial learning methods and datasets (e.g., a synthetic dataset and CI-MNIST) without fairness implications and uses dp and eodd as the fairness metrics. In contrast, our benchmark is more comprehensive in terms of algorithms, datasets, and fairness evaluation metrics.

## B    DETAILS OF THE GROUP FAIRNESS

In this section, we provide the details of the group fairness. We first introduce the definition of group fairness. Then, we introduce the existing group fairness metrics and algorithms.

- dp (Demographic Parity or Statistical Parity) (Zemel et al., 2013). A classifier satisfies demographic parity if the predicted outcome $\hat{Y}$ is independent of the sensitive attribute $S$, i.e., $P(\hat{Y} \mid S = 0) = P(\hat{Y} \mid S = 1)$.

- prule (Zafar et al., 2017). A classifier satisfies $p\%$-rule if the ratio between the probability of subjects having a certain sensitive attribute value assigned the positive decision outcome and the probability of subjects not having that value also assigned the positive outcome should be no less than $p$/100, i.e., $|P(\hat{Y} = 1 \mid S = 1)/P(\hat{Y} = 1 \mid S = 0)| \leq p/100$.

- eopp (Equality of Opportunity) (Hardt et al., 2016). A classifier satisfies equalized opportunity if the predicted outcome $Y$ is independent of the sensitive attribute $S$ when the label $Y = 1$, i.e., $P(\hat{Y} \mid S = 0, Y = 1) = P(\hat{Y} \mid S = 1, Y = 1)$.

- eodd (Equalized Odds) (Hardt et al., 2016). A classifier satisfies equalized odds if the predicted outcome $Y$ is independent of the sensitive attribute $S$ conditioned on the label $Y$, i.e., $P(\hat{Y} \mid S = 0, Y = y) = P(\hat{Y} \mid S = 1, Y = y), y \in \{0, 1\}$.

- acc (Accuracy Parity). A classifier satisfies accuracy parity if the error rates of different sensitive attribute values are the same, i.e., $P(\hat{Y} \neq Y \mid S = 0) = P(\hat{Y} \neq Y \mid S = 1), y \in \{0, 1\}$.

- aucp (ROC AUC Parity). A classifier satisfies ROC AUC parity if its area under the receiver operating characteristic curve of w.r.t. different sensitive attribute values are the same.

- ppv (Predictive Parity Value Parity) A classifier satisfies predictive parity value parity if the probability of a subject with a positive predictive value belonging to the positive class w.r.t. different sensitive attribute values are the same, i.e., $P(Y = 1 \mid \hat{Y}, S = 0) = P(Y = 1 \mid \hat{Y}, S = 1)$.

- bnegc (Balance for Negative Class). A classifier satisfies balance for the negative class if the average predicted probability of a subject belonging to the negative class is the same w.r.t. different sensitive attribute values, i.e., $\mathbb{E}[f(X) \mid Y = 0, S = 0] = \mathbb{E}[f(X) \mid Y = 0, S = 1]$.

- bposc (Balance for Positive Class). A classifier satisfies balance for the negative class if the average predicted probability of a subject belonging to the positive class is the same w.r.t. different sensitive attribute values, i.e., $\mathbb{E}[f(X) \mid Y = 1, S = 0] = \mathbb{E}[f(X) \mid Y = 1, S = 1]$.

- abcc (Area Between Cumulative density function Curves) (Han et al., 2023) is proposed to precisely measure the violation of demographic parity at the distribution level. The new fairness metrics directly measure the difference between the distributions of the prediction probability for different demographic groups

## C    EXPERIMENTAL SETTING

For tabular datasets, we use a two-layer Multi-layer Perceptron with 256 neurons each for all datasets. We use different bath sizes for different datasets based on the total number of instances of each dataset. For image datasets, we use various neural networks (such as ResNet-18 (He et al., 2016) and ResNet-152) as the backbone. We don't use weight decay for all datasets and all fairness methods. We use linear learning rate decay for all datasets and all fairness methods. We use Adam (Diederik P. Kingma, 2014) as the optimizer with a learning rate of 0.001 for both tabular and image data. As these objectives, utility and fairness, often present trade-offs, it can be challenging to determine when to stop model training, and this issue is rarely discussed in the previous literature. In this work, we adopted a straightforward stopping strategy *based on our experience*. We employ a linear decay strategy for the learning rate, halving it every 50 training steps. The model training is stopped when the learning rate decreases to a value below $1e^{-5}$.

## D  DETAILS OF THE BENCHMARKING DATASETS

In this section, we provide the details of the benchmarking datasets. We first introduce the benchmarking datasets. Then, we introduce the data preprocessing and data splitting.

- **Tabular Datasets**

  - `Adult`[3] (Kohavi & Becker, 1996). The Adult dataset is widely used in machine learning and data mining research. It contains 1994 U.S. census data. The task of the dataset is to predict whether a person makes over $50K a year, given an individual's demographic and financial information. The dataset includes sensitive information such as age and gender. In the literature, gender is mostly used as the (binary) sensitive attribute to evaluate group fairness.

  - `COMPAS`[4] (Larson et al., 2016). The COMPAS dataset contains records of criminal defendants, and it is used to predict whether the defendant will recidivate within two years. The dataset includes attributes related to the defendant, such as their criminal history, and demographic information, such as gender and race.

  - `German`[5] (Dua & Graff, 2017). The German Credit dataset contains information on individuals who applied for credit at a German bank, including their financial status, credit history, and demographic information (e.g., gender and age). It is used to predict whether an individual should receive a positive or negative credit risk rating.

  - `Bank`[6] (Dua & Graff, 2017). The bank marketing dataset is used to analyze the effectiveness of marketing strategies of a Portuguese banking institution by predicting if the client will subscribe to a term deposit. The input variables of the dataset include the bank client's personal information and other bank marketing activities related to the client. Age was studied as the sensitive attribute in (Zafar et al., 2017).

  - `KDDCensus`[7] (Dua & Graff, 2017). Similar to the Adult dataset, the task of the KDD Census dataset is to predict whether the individual's income is above $50k with more instances. The sensitive attributes are gender and race.

  - `ACS-I/E/P/M/T`[8] (Ding et al., 2021). The ACS dataset provides several prediction tasks (e.g., predict whether an individual's income is above $50K or whether an individual is employed). It is constructed from the American Community Survey (ACS) Public Use Microdata Sample (PUMS). All tasks contain features for race, gender, and other task-related features.

- **Image Datasets**

  - `CelebA-A/W/S`[9] (Liu et al., 2015). The CelebFaces Attributes dataset comprises 20k face images from 10k celebrities. Each image is annotated with 40 binary labels indicating specific facial attributes such as gender, hair color, and age.

  - `UTKFace`[10] (Zhang et al., 2017). The UTKFace dataset is a large-scale face dataset that contains over 20k face images of people from different ethnicities and ages. The images are annotated with age, gender, and ethnicity information.

- **Text Datasets**

  - `Jigsaw`[11] (Jigsaw, 2018). The CelebFaces Attributes dataset comprises 20k face images from 10k celebrities. Each image is annotated with 40 binary labels indicating specific facial attributes such as gender, hair color, and age.

---

[3]https://archive.ics.uci.edu/ml/datasets/adult
[4]https://github.com/propublica/compas-analysis
[5]https://archive.ics.uci.edu/dataset/144/statlog+german+credit+data
[6]https://archive.ics.uci.edu/dataset/222/bank+marketing
[7]https://archive.ics.uci.edu/ml/datasets/Census-Income+(KDD)
[8]https://github.com/zykls/folktables
[9]https://mmlab.ie.cuhk.edu.hk/projects/CelebA.html
[10]https://susanqq.github.io/UTKFace/
[11]https://www.kaggle.com/c/jigsaw-toxic-comment-classification-challenge

# E    DETAILED EXPERIMENTAL SETTINGS

We present the details of the experimental setting in Tables 5 to 7. Table 5 presents the common hyperparameters used by both Tabular and Image datasets, including an initial learning rate of 0.01, Adam as the optimizer, zero weight decay, StepLR as the scheduler with a step of 50, a gamma value of 0.1, and 150 training steps in total. Table 6 presents the range of control hyperparameters used for various fairness methods. Each method has a unique range of these parameters. Table 7 indicates the batch sizes chosen for various datasets during training, ranging from 32 for the German and COMPAS datasets to a large 4096 for the KDDCensus and ACS-I/E/P/M/T datasets, with CelebA-A/W/S and UTKFace datasets using a batch size of 128, which are determined by the number of instances of the datasets.

Table 5: Common Hyper-parameters.

| Dataset | Initial LR | Optimizer | Weight Decay | Scheduler | StepLR_step | StepLR_gamma | Training steps |
|---------|-----------|-----------|--------------|-----------|-------------|--------------|----------------|
| Tabular | 0.01 | Adam | 0.0 | StepLR | 50 | 0.1 | 150 |
| Image | 0.01 | Adam | 0.0 | StepLR | 50 | 0.1 | 150 |

Table 6: The fairness control hyperparameter selections.

| Dataset | Control hyperparameter |
|---------|------------------------|
| DiffDP | $0.2, 0.4, 0.6, 0.8, 1.0, 1.2, 1.4, 1.6, 1.8, 2.0, 2.5, 3.0, 3.5, 4$ |
| DiffEodd | $0.2, 0.4, 0.6, 0.8, 1.0, 1.2, 1.4, 1.6, 1.8, 2.0, 2.5, 3.0, 3.5, 4$ |
| DiffEopp | $0.2, 0.4, 0.6, 0.8, 1.0, 1.2, 1.4, 1.6, 1.8, 2.0, 2.5, 3.0, 3.5, 4$ |
| PRemover | $0.05, 0.1, 0.15, 0.2, 0.25, 0.3, 0.35, 0.40, 0.45, 0.50, 0.6, 0.7, 0.8, 0.9, 1.0$ |
| HSIC | $50, 100, 150, 200, 250, 300, 350, 400, 450, 500, 600, 700, 800, 900, 1000$ |
| AdvDebias | $0.2, 0.4, 0.6, 0.8, 1.0, 1.2, 1.4, 1.6, 1.8, 2.0, 2.5, 3.0, 3.5, 4$ |
| LAFTR | $0.1, 0.2, 0.3, 0.4, 0.5, 1, 2, 3, 4, 5$ |

Table 7: The batch size for different datasets during the training.

| Dataset | Bank | German | Adult | COMPAS | KDDCensus | ACS-I/E/P/M/T | CelebA-A/W/S | UTKFace |
|---------|------|--------|-------|--------|-----------|---------------|--------------|---------|
| Batch Size | 1024 | 32 | 1024 | 32 | 4096 | 4096 | 128 | 128 |

# F MORE EXPERIMENT RESULTS ON Adult

In this appendix, we present the experimental results on Adult datasets. The same experiment results are presented in http://TODO.

## F.1 UTILITY-FAIRNESS TRADE-OFFS

We plot the utility-fairness trade-offs for the Adult dataset with gender as the sensitive attribute and present the results in Figures 7 and 8.

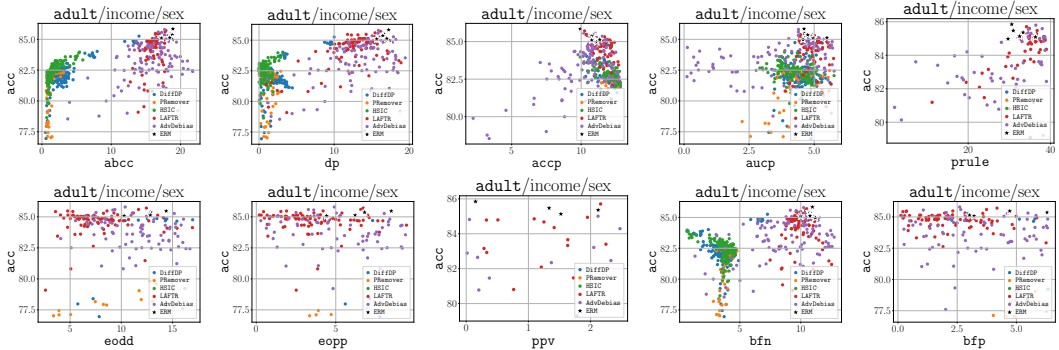

Figure 7: The Utility-Fairness Trade-offs with acc as utility metric.

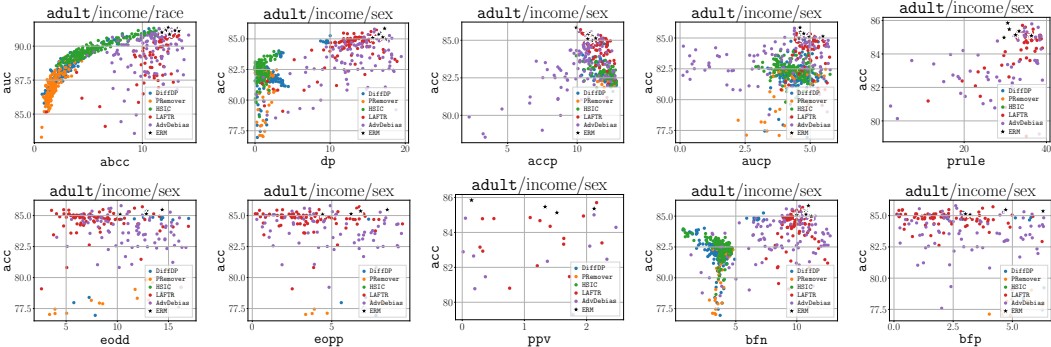

Figure 8: The Utility-Fairness Trade-offs with acc as utility metric.

## F.2 TRAINING CURVES AND HYPERPARAMETERS FOR CONTROLLING FAIRNESS

We plot the utility and fairness training curves for varying fairness control hyperparameters on the Adult dataset, and present the results in Figure 9. The intensity of the color represents the size of the control parameters. In most cases, the **larger** value of control parameters yields better fairness performance, while **small** ones have worse fairness performance.

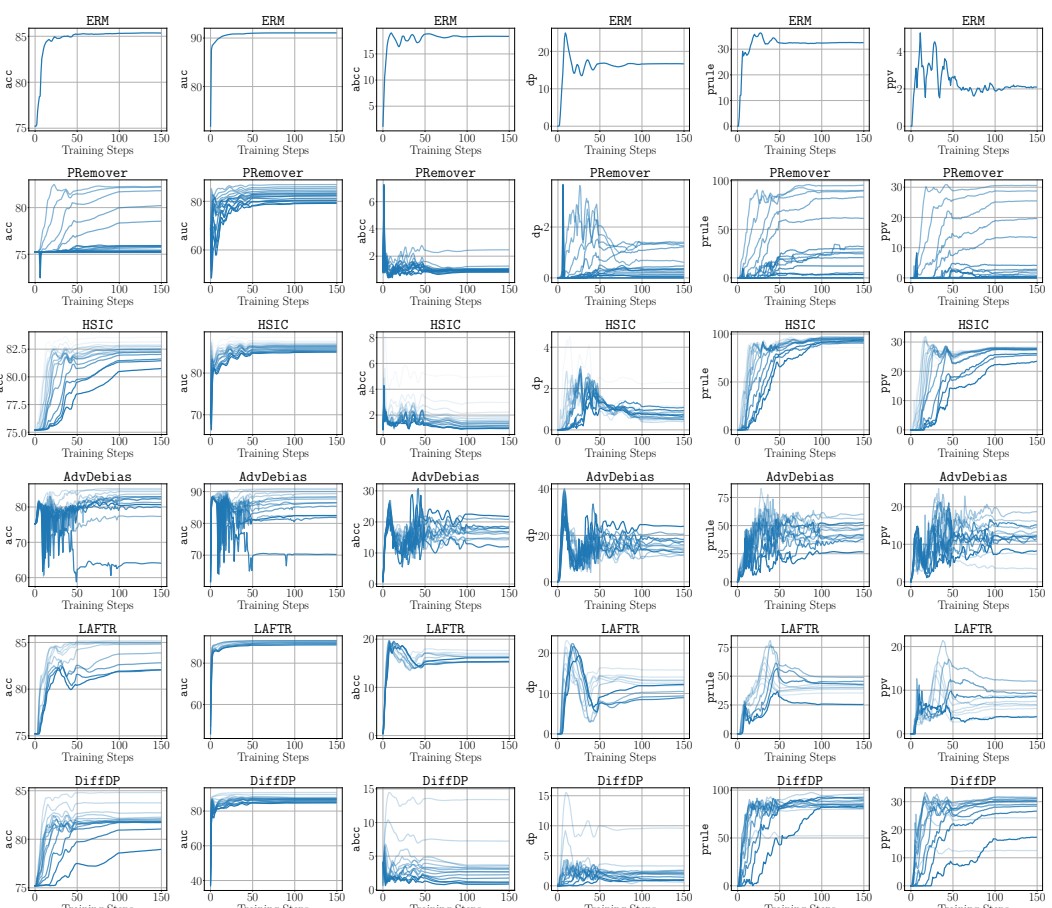

Figure 9: Hyperparameters for Controlling Fairness on Adult dataset.

## G    MORE EXPERIMENT RESULTS ON CelebA-A

In this appendix, we present the experimental results on the CelebA-A dataset.

### G.1    UTILITY-FAIRNESS TRADE-OFFS

We plot the utility-fairness trade-offs for the CelebA-A dataset with gender as the sensitive attribute and present the results in Figure 10. The results show the utility-fairness trade-offs in CelebA-A dataset.

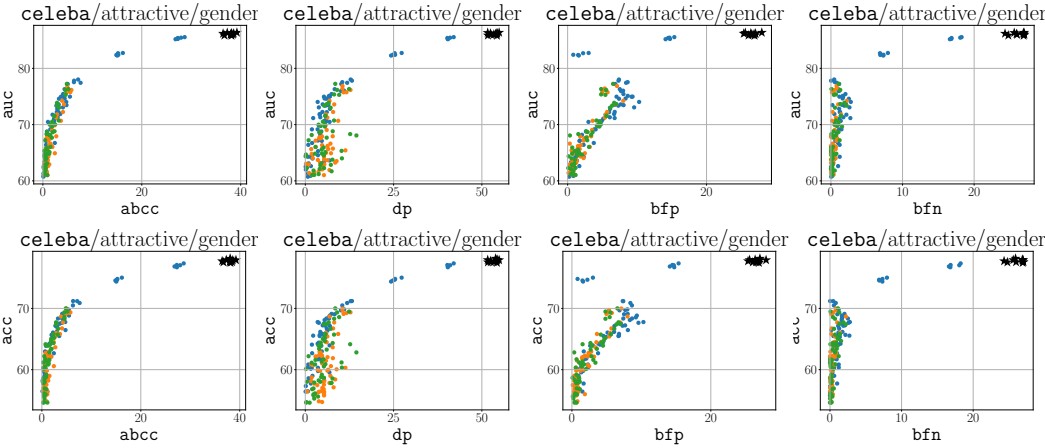

Figure 10: The Utility-Fairness Trade-offs

### G.2    TRAINING CURVES AND HYPERPARAMETERS FOR CONTROLLING FAIRNESS

We plot the utility and fairness training curves for varying fairness control hyperparameters on the CelebA-A dataset, and present the results in Figure 11. The intensity of the color represents the size of the control parameters. In most cases, the **larger** value of control parameters yields better fairness performance, while small ones have worse fairness performance.

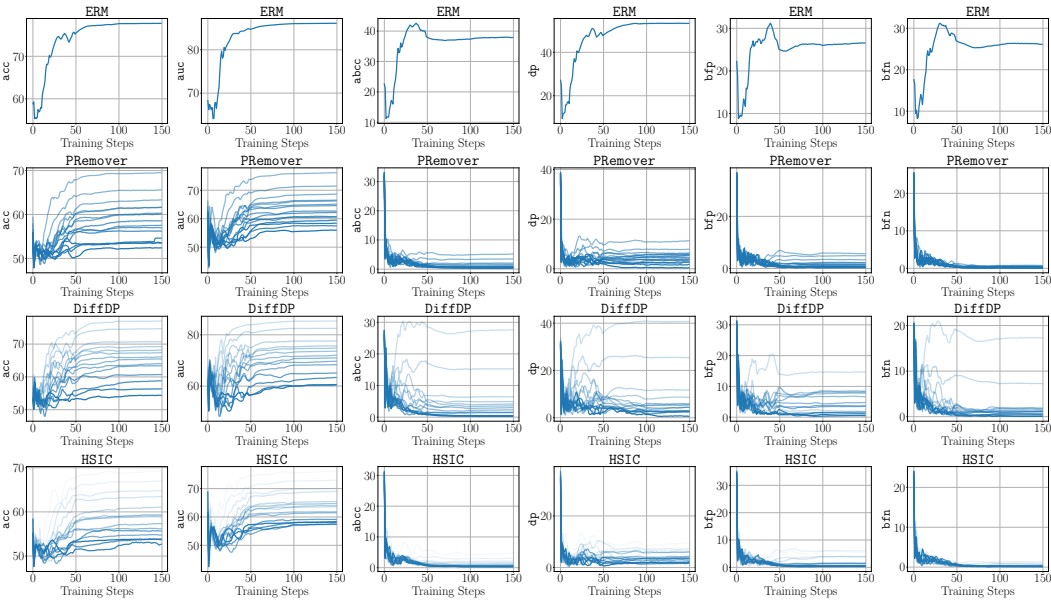

Figure 11: Training Curves and Hyperparameters for Controlling Fairness one CelebA-A dataset.

# H MORE EXPERIMENT RESULTS ON Jigsaw

We present the results on fairness performance using various fairness control hyperparameters. We observed that the fairness metrics abcc and dp cannot be influenced by the fairness control hyperparameters, whereas prule can.

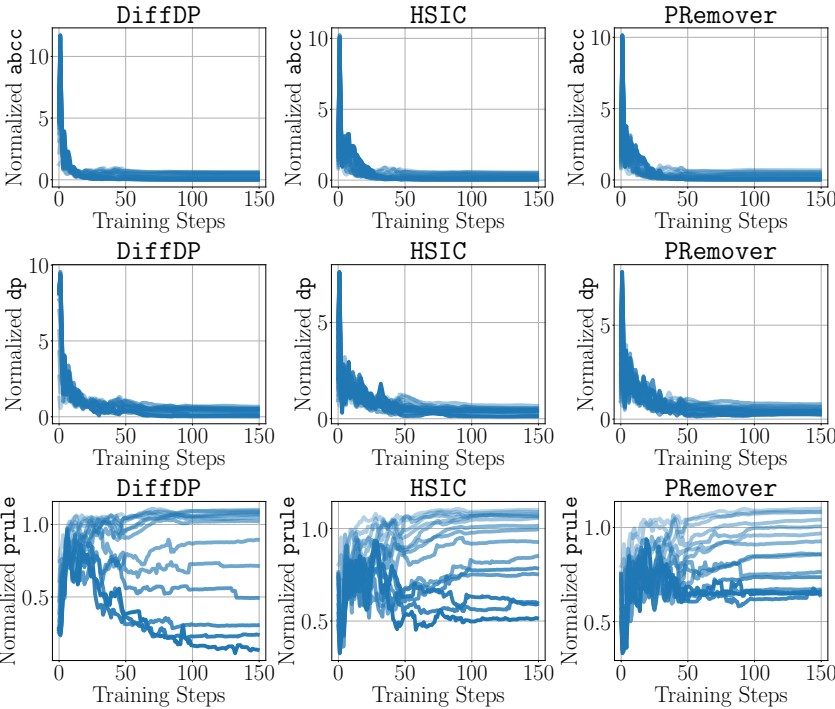

Figure 12: The fairness performance with varying fairness control hyperparameters on text data. The intensity of the color represents the size of the control parameters. In most cases, the **larger** value of control parameters yields better fairness performance, while small ones have worse fairness performance.

## H.1 HOW DOES MODEL SIZE INFLUENCE FAIRNESS PERFORMANCE?

We conducted experiments to explore the influence of model size on fairness performance. We use various neural networks with the number of neural network trainable parameters spanning from 11.6M to 126.9M.[12] The results are presented in Figure 13.

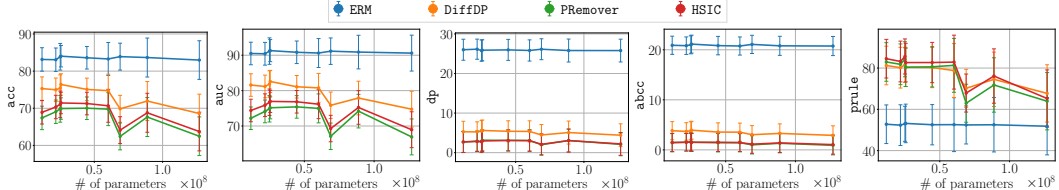

Figure 13: The performance with varying sizes of neural networks. The x-axis is the number of model parameters. There is no clear relationship between model size and performance.

⑧ **The architecture does not significantly influence fairness performance.** An increase in neural network parameters does not yield significant changes in utility or fairness performance. This suggests larger models do not naturally mitigate that dataset bias. The results observed from the `DiffDP` method indicate a potential correlation between utility performance and bias: fairness performance may deteriorate as utility increases, exhibiting trade-offs between them across different models.

---

[12]ResNet-18 (11.6M), ResNet-34 (21.8M), ResNet-50 (25.6M), ResNet-101 (44.5M), ResNet-152 (60.2M), ResNext-50 (25.0M), ResNext101 (88.8M), wide_ResNet-50 (68.9M), and wide_ResNet101 (126.9M).

# I    IMPLEMENTATION COMPARISON WITH AIF360 AND FAIRLEARN

In this section, we provide the implementations of adversarial debiasing in AIF360, FairLearn, and FFB to demonstrate FFB are extensible, minimalistic, and research-oriented compared to existing fairness packages. Algorithm 1, Algorithm 2, and Algorithm 3 show the implementation of adversarial debiasing in AIF360, FairLearn, and FFB, respectively.

We can see that both AIF360 and FairLearn use Scikit-learn API design (e.g., the use of `fit()` function), whereas FFB use Pytorch-style implementation, which provides a unified data preprocessing pipeline and fairness evaluation interface in a single script. Thus, researchers using FFB can use the bias mitigation method and reproduce the experimental results using one line of command. Additionally, AIF360 and FairLearn use complicated class inheritance (e.g., `AdversarialFairnessClassifier` in FairLearn inherents `AdversarialFairness` and `ClassifierMixin`), `AdversarialDebiasing` in AIF360 inherents `Transformer`), and other external dependencies (e.g., `AdversarialFairness` in FairLearn uses `backendEngine_` to implement the training step), making the implementation hard to read. This makes researchers hard to understand and re-implement the bias mitigation methods.

---

**Algorithm 1** `AdvDebias` in AIF360

```python
class AdversarialDebiasing(Transformer):

    def __init__(self):
        ...

    def _classifier_model(self, features, features_dim, keep_prob):
        ... # deine classifier

    def _adversary_model(self, pred_logits, true_labels):
        ... # deine adversary model

    def predict(self, dataset):
        ...

    def fit(self, dataset):

        with tf.variable_scope(self.scope_name):

            # tf graph construction
            ...

            self.sess.run(tf.global_variables_initializer())
            self.sess.run(tf.local_variables_initializer())

            for epoch in range(self.num_epochs):
                # training
                for i in range(num_train_samples//self.batch_size):
                    batch_feed_dict = {self.features_ph: batch_features,
                                       self.true_labels_ph: batch_labels,
                                       self.protected_attributes_ph: batch_protected_attributes,
                                       self.keep_prob: 0.8}
                    if self.debias:
                        _, _, pred_labels_loss_value, pred_protected_attributes_loss_vale = self.sess.run([
                                       classifier_minimizer,
                                       adversary_minimizer,
                                       pred_labels_loss,
                                       pred_protected_attributes_loss], feed_dict=batch_feed_dict)
                        if i % 200 == 0:
                            ... # logging
                    else:
                        _, pred_labels_loss_value = self.sess.run(
                            [classifier_minimizer,
                             pred_labels_loss], feed_dict=batch_feed_dict)
                        if i % 200 == 0:
                            ... # logging
```

---

**Algorithm 2** AdvDebias in FairLearn

```python
class _AdversarialFairness(BaseEstimator):

    def __init__(self):
        ...

    def __setup(self, X, Y, A):
        ...

    def fit(self, X, y, *, sensitive_features=None):
        X, Y, A = self._validate_input(X, y, sensitive_features, reinitialize=True)

        ...

        for epoch in range(epochs):
            batch_slice = slice(
                    batch * batch_size,
                    min((batch + 1) * batch_size, X.shape[0]),
                )
            (LP, LA) = self.backendEngine_.train_step(
                X[batch_slice], Y[batch_slice], A[batch_slice]
            )
            predictor_losses.append(LP)
            adversary_losses.append(LA)

            ...

    def partial_fit(self, X, y, *, sensitive_features=None):
        ...

    def decision_function(self, X):
        ...

    def predict(self, X):
        ...

    def _validate_input(self, X, Y, A, reinitialize=False):
        ...

    def _validate_backend(self):
        ...

    def _set_predictor_function(self):
        ...

class AdversarialFairnessClassifier(_AdversarialFairness, ClassifierMixin):
    def __init__(self):
        """Initialize model by setting the predictor loss and function."""
        self._estimator_type = "classifier"
        super(AdversarialFairnessClassifier, self).__init__()
```

**Algorithm 3** AdvDebias in FFB

```python
class Adversary(nn.Module):
    ...

class MLP(nn.Module):
    ...

def test(model, test_loader, criterion, device, args=None):
    ...

def train(clf, adv, data_loader, clf_criterion, adv_criterion, clf_optimizer, adv_optimizer):
    ...

if __name__ == '__main__':
    parser = argparse.ArgumentParser()
    parser.add_argument('--dataset', type=str, default="adult")
    parser.add_argument('--model', type=str, default="MLP")
    ...
    args = parser.parse_args()

    # Dataset selection
    if args.dataset == "adult":
        X, y, s = load_adult_data(sensitive_attribute=args.sensitive_attr)
    elif args.dataset == "compas":
        X, y, s = load_compas_data( sensitive_attribute=args.sensitive_attr)
    ...

    # Unified Dataset preprocessing (e.g., train/test split, )
    ...

    # define network architecture, etc. optimizer
    clf = MLP(n_features=n_features, num_classes=1, mlp_layers=[512, 256, 64]).to(device)
    clf_criterion = nn.BCELoss()
    clf_optimizer = optim.Adam( clf.parameters(), lr=args.lr)

    adv = Adversary( n_sensitive = 1 ).to(device)
    adv_criterion = nn.BCELoss(reduction="mean")
    adv_optimizer = optim.Adam(adv.parameters(), lr=args.lr)

    for epoch in range(1, args.num_epochs+1):
        # begin training
        train(clf, adv, train_loader, clf_criterion, adv_criterion, clf_optimizer, adv_optimizer)

        if epoch % args.logging_steps == 0 or epoch == args.num_epochs:
            test_metrics = test(model=clf, test_loader=test_loader, criterion=clf_criterion, device=device)
            # logging metrics
            ...
```

