# OpenReview forum: "FFB: A Fair Fairness Benchmark for In-Processing Group Fairness Methods"
_ICLR.cc/2024/Conference — ICLR 2024 poster_

### Official Review · Reviewer_8vvg · 2023-10-12

**Soundness:** 3 good
**Presentation:** 3 good
**Contribution:** 3 good
**Rating:** 8
**Confidence:** 4

**Summary:**

Paper introduces the Fair Fairness Benchmark, a benchmarking framework for in-processing group fairness methods.
Contributions are: the provision of flexible, extensible, minimalistic, and research-oriented open-source code;
the establishment of unified fairness method benchmarking pipelines.

**Strengths:**

Good amount of fairness metrics.
Good amount of datasets.

**Weaknesses:**

I think that the main missing point of the paper is a larger series of state of the art (un)fairness mitigation methods to use as baseline.

**Questions:**

I think that authors should elaborate on the limited amount of of state of the art (un)fairness mitigation methods.

---

> ### Author Response · Authors · 2023-11-21
> **Response to Reviewer 8vvg [key points: explain the scope of our work]**
>
> Dear Reviewer 8vvg,
>
> We sincerely thank you for your time and effort in reviewing our paper and your positive support of our paper. We address your concerns point by point in the following:
>
>
> **Q: I think that authors should elaborate on the limited amount of of state of the art (un)fairness mitigation methods**
>
>
> Thank you for your valuable feedback on our paper. We appreciate your suggestion to include a broader array of methods for mitigating unfairness. Indeed, diversifying the techniques we explore could add depth to our research. In the following, we would like to clearfy the decision to focus on the most commonly used baseline fairness mitigation methods.
>
> 1. **Benchmarking most commonly used methods:** Our choice to concentrate on most commonly used methods was driven by a clear intent: to establish a solid foundation for fairness. By employing these well-accepted baseline methods, we not only adhere to established standards but also provide a reliable benchmark. This approach ensures that our findings are grounded in the field’s current understanding and opens avenues for direct comparison with existing research.
>
> 2. **In-Depth Analysis:** Our experimental design was not a mere surface-level examination on numerous methods. We conducted extensive experiments with these chosen methods to ensure a comprehensive and nuanced understanding. Our investigation delved into the intricacies and outcomes of these established techniques, offering a detailed and rigorous examination of their effectiveness and limitations across various scenarios. This depth of analysis contributes significantly to the body of knowledge.
>
>
>
> We thank you for your support of our paper again and hope that our response adequately addresses your concerns.
>
> Thanks,\
> Authors

---

### Official Review · Reviewer_Tj8b · 2023-10-17

**Soundness:** 3 good
**Presentation:** 3 good
**Contribution:** 2 fair
**Rating:** 5
**Confidence:** 4

**Summary:**

This paper proposes a group fairness benchmark for in-processing methods. A wide range of fairness definitions are used along with multiple datasets. Three types of in-processing methods are compared: gap regularization, independence, and adversarial learning. Several observations are made on fairness-utility tradeoff, stability, and others.

**Strengths:**

* Since the fairness literature is vast, it is a good time to make a comprehensive comparison.
* The comparison of in-processing methods using group fairness measures looks reasonable.

**Weaknesses:**

* While proposing a fairness benchmark is a worthy effort, the scope of this study seems a bit limited because it only considers in-processing methods. Even AIF360 proposed in 2018 compares pre-processing, in-processing, and post-processing methods, so it is expected that a new benchmark should at least subsume this scope. Pre-processing methods should not be ignored because some of them are designed to complement in-processing methods for the best results. Even for in-processing methods only, there are reweighing and sampling methods [1,2,3] that should be compared.

Nowadays, performing in-processing training without sensitive attributes is also actively studied, and a comparison with these methods (e.g., [4,5]) would be interesting.

[1] Jiang et al., Identifying and Correcting Label Bias in Machine Learning, AISTATS 2020
[2] Roh et al., FairBatch: Batch Selection for Model Fairness, ICLR 2021
[3] Iosifidis and Ntoutsi, AdaFair: Cumulative Fairness Adaptive Boosting, CIKM, 2019
[4] Lahoti et al., Fairness without Demographics through Adversarially Reweighted Learning, NeurIPS 2020
[5] Hashimoto et al., Fairness Without Demographics in Repeated Loss Minimization, ICML 2018

* Concluding that HSIC is the best approach seems misleading because not all in-processing methods were compared as explained above.

* Some of the key observations are already known in the fairness community. Observation 4 (adversarial debiasing has instability) is not surprising and is mentioned in the papers that use this approach. Observation 5 (utility-fairness trade-off is controllable) does not seem revealing either. What's actively studied nowadays is whether there has to be a trade-off or not, and there is a line of research that discusses when utility and fairness align instead of conflict. It would be interesting to empirically verify if the claims made here are actually true. In observation 6 (training curve stability), a future direction is suggested to "focus on enhancing fairness training stability". However, it is not clear why enhancing the fairness stability is the most important research direction among other challenges. What does it mean to be not stable enough?

* There is an emphasis on making the benchmark research-oriented, but this term is rather vague. Instead, benchmarks should target practical applications as they actually show which fairness method works.

* One of the future works is to include a wider range of in-processing group fairness methods. This direction should not be a future work as the current paper claims to be a complete benchmark for such methods.

**Questions:**

Please address the weak points above.

---

> ### Author Response · Authors · 2023-11-21
> **[1/2] Repsonse to Reviewer Tj8b [key points: explain the scope of our work; adjust the observation accordingly]**
>
> Dear Reviewer Tj8b,
>
> We sincerely thank you for your time and effort in reviewing our paper. Your comments that 'a good time to make a comprehensive comparison' and 'reasonable comparison of in-processing methods' are highly appreciated and encouraging.  Most importantly, we value your suggestions and address them point by point in the following:
>
>
> **Q1: scope of this study seems a bit limited because it only considers in-processing methods..... Nowadays, performing in-processing training without sensitive attributes is also actively studied, and a comparison with these methods would be interesting. ......One of the future works is to include a wider range of in-processing group fairness methods. This direction should not be a future work as the current paper claims to be a complete benchmark for such methods.**
>
> *---we combine your three concerns about the number of benchmarking methods in this question.---*
>
> We thank you for your insightful and valuable suggestion. We agree that incorporating more settings would result in a more comprehensive fairness benchmark. This includes integrating more fairness algorithms at different stages (pre-processing, post-processing), expanding the definitions of fairness (individual fairness, counterfactual fairness), and performing in-processing training without sensitive attributes.
>
> In the following, we would like to clarify the decision to focus on the most commonly used baseline fairness mitigation methods.
>
> 1. **Benchmarking most commonly used methods:** Our choice to concentrate on the most commonly used methods was driven by a clear intent: to establish a solid foundation for fairness. By employing these well-accepted baseline methods, we not only adhere to established standards but also provide a reliable benchmark. This approach ensures that our findings are grounded in the field’s current understanding and opens avenues for direct comparison with existing research.
>
> 2. **In-Depth Analysis:** Our experimental design was not a mere surface-level examination of numerous methods. We conducted extensive experiments with these chosen methods to ensure a comprehensive and nuanced understanding. Our investigation delved into the intricacies and outcomes of these established techniques, offering a detailed and rigorous examination of their effectiveness and limitations across various scenarios. This depth of analysis contributes significantly to the body of knowledge.
>
>
>
>
> **Q2: Concluding that HSIC is the best approach seems misleading because not all in-processing methods were compared as explained above.**
>
> We thank you for your insightful suggestion. We have modified this observation to "The HSIC achieves the best utility-fairness trade-off among the tested methods".
>
>
> **Q3: Some of the key observations are already known in the fairness community.**
>
> In our paper, we acknowledge that several observations made might not be entirely new. We have carefully reviewed and removed some observations that are well-established in the field. Instead, our focus has shifted to highlighting those findings that have not been explicitly detailed in previous work as follows:
> - ➀ Not all widely used fairness datasets stably exhibit fairness issues.
> - ➁ Utility-fairness trade-offs are generally controllable by the hyperparameter.
> - ➂ The HSIC achieves the best utility-fairness trade-off among the tested methods.
> - ➃ Stopping training while learning rate is lower enough is effective.
>
> We also modified our paper about the findings from our experiments on Page 2.

---

> > ### Author Response · Authors · 2023-11-21
> > **[2/2] Repsonse to Reviewer Tj8b [key points: explain the scope of our work; adjust the observation accordingly]**
> >
> > **Q4: There is an emphasis on making the benchmark research-oriented, but this term is rather vague.**
> >
> > We would like to clearfy the "research-oriented". The term 'research-oriented' exemplifies our philosophy in this work: that code for research should be both simple and free from redundant elements. In this work, we firmly adhere to and follow this philosophy. This guiding principle also significantly shapes and differentiates FFB.
> >
> > - **Emphasis on Code Simplicity**: The FFB embodies our philosophy that simplicity is paramount in research code. Its design is deliberately straightforward, including the dataset loading function and metric functions, which can be easily understood without documentation. This simplicity not only facilitates modifications and extensions but also promotes seamless integration into one's own research.
> >
> > - **Facilitation of Research Objectives**: Adhering to these principles, FFB effectively furthers our goal of making research more accessible and focused. Researchers can concentrate on their primary objectives, avoiding complex or redundant code.
> >
> >
> > Following the above two, we achieve our research goal that
> > - **Comprehensive Evaluation of Commonly Used Fairness Methods**: We provide extensive experimental evidence to support previous observations, such as the instability of adversarial training and the utility-fairness trade-off. In addition, we also make new findings, including the presence of fairness in widely-used datasets and strategies for determining training stop.
> >
> >
> >
> > We hope our response adequately addresses your concerns. If so, we kindly request the reviewer to re-evaluate our work and consider raising their score.
> >
> >
> > Thanks,\
> > Authors

---

> ### Author Response · Authors · 2023-11-22
> **Thanks for raising your score**
>
> Dear Reviewer Tj8b,
>
> We thank you for raising your score and are more than happy to answer any questions you may have.
>
> Thank you again.
>
> Sincerely,\
> Authors

---

### Official Review · Reviewer_dSax · 2023-10-19

**Soundness:** 3 good
**Presentation:** 3 good
**Contribution:** 3 good
**Rating:** 8
**Confidence:** 4

**Summary:**

The paper introduces the Fair Fairness Benchmark (FFB), a framework for evaluating group fairness methods in machine learning. It aims to address the challenges of inconsistent experimental settings, limited algorithmic implementations, and extensibility issues in fairness tools. FFB offers an open-source benchmark, standardized code, and extensive analysis from 45,079 experiments, making it a valuable resource for the fairness research community.

**Strengths:**

The paper address a prominent problem in the Fairness community which is the inconsistency of different results from various papers. A lot of experiments are conducted in a standardized manner using different datasets, methods and evaluation metrics. The writing is clear and linking the contribution points with 1,2,3.. in the text makes it quite convenient to read. The open source code looks good and is in a state which can be easily adopted for other researchers.

**Weaknesses:**

In Table 4 you give some recommendations if one should use the dataset or not. In the text you explain the different discussion. It can be that I missed it but are there any quantifiable measurements to check if a dataset is good for fairness or not - like a metric? And if so can this value be included in the study? Also you said you had 10 trials to produce this table. Did you do any HP optimization with some hold-out splits or how exactly was this done? More detailed information about this would be appreciated.

**Questions:**

Figure 1: Are the results different because of different random seeds or did you changed the gamma values of the loss functions to obtain different acc-fairness trade-offs?

**Details Of Ethics Concerns:**

This study has a potential positive influence on the evaluation of fair algorithms. However there are also risks involved if the benchmark is not carefully chosen. Maybe some of this risks (stakeholders how chose the benchmark, what metric is weighted how much, have discriminated people the chance to be involved in the process, etc.) can be discussed in the paper.

---

> ### Author Response · Authors · 2023-11-21
> **Response to Reviewer dSax [key points: provide details about Table 4 and Figure 1]**
>
> Dear Reviewer dSax,
>
> We are genuinely grateful for the time and effort you devoted to reviewing our paper. Your acknowledgment of our work addressing a prominent problem in the Fairness community is highly appreciated. We are also pleased to hear that our writing is clear and our contribution is clear. Additionally, we are glad that our open-source code meets your standards and is easily adaptable for fellow researchers. Most importantly, we highly value your suggestions and have addressed each of them in detail in the following:
>
>
> **Q1: More detailed information about Table 4 this would be appreciated.**
>
> In Table 4, we offer recommendations on whether to use specific datasets for fairness, based on qualitative assessments rather than quantifiable measurements. Although we recognize the importance of the measurements, our current methodology focuses mainly on evaluating the numerical values of fairness metrics to form our recommendations. Regarding the experimental procedure, we conducted ten trials using the same neural network architecture but with varying data splits.
>
> **Q2: Figure 1: Are the results different because of different random seeds or did you change the gamma values of the loss functions to obtain different acc-fairness trade-offs?**
>
> In our experiments, we varied both the random seeds and the gamma values of the loss functions, which contributed to different accuracy-fairness trade-offs. Using all the raw results with various random seeds and gamma values ensures a comprehensive and realistic presentation of the results. The substantial number of runs (27,568) represented in these figures guarantees the reliability of our findings.
>
>
> We thank you for your support of our paper and hope that our response adequately addresses your concerns.
>
> Thanks,\
> Authors

---

### Official Review · Reviewer_5ZmP · 2023-11-03

**Soundness:** 2 fair
**Presentation:** 4 excellent
**Contribution:** 2 fair
**Rating:** 6
**Confidence:** 4

**Summary:**

This paper proposes a new fairness benchmark called FFB (Fair Fairness Benchmark). FFB targets to support group fairness metrics and in-processing fairness algorithms, and the system contains several well-known fairness algorithms with metrics. The paper also describes various observations, which are gathered by using the proposed benchmark. For example, the paper observes that the model architecture usually does not significantly affect the fairness performances, which shows that the biases are mainly from the training data. These observations are aggregated based on more than 45000 experiments.

**Strengths:**

- The paper aims to solve a very important problem in the fairness literature, the lack of great benchmarks.
- The paper well states the challenges in making fairness benchmarks and proposes a new one called FFB to help the fairness literature.
- The paper performs extensive experiments and summarizes their observations in several aspects, including model performance and stability.

**Weaknesses:**

Although I appreciate the paper’s contribution on proposing a new benchmark, I have several concerns on the manuscript as a research paper.
- The paper needs to explain more clearly how to use FFB and the strengths of the system itself.
  - The paper does not clearly explain how to use FFB for testing new algorithms or new datasets. The paper currently focuses on the predefined algorithms and models in FFB, but as a paper that proposes a new benchmark, demonstrating how to utilize their system can be more important.
  - Similarly, it would be better if the paper could provide more explanations on the strengths of FFB itself. It seems some explained characteristics (like minimalistic aspect) are not supported by enough convincing explanations.
- Currently, the paper explains their observations on several algorithms and datasets by using FFB, but many of the observations are not very surprising and already discussed in the literature. Thus, it is a bit unclear to me whether such observations themselves can be a strong contribution of this paper. Although such observations are still noteworthy to summarize in the paper, it may be better to not oversell them in the paper. It would be better if the paper could clearly connect these observations to the previous knowledge in the fairness literature.

**Questions:**

My main concerns are described in the above weakness section. I hope to hear the authors’ response to them.

--------------------
The score is updated after rebuttal.

---

> ### Author Response · Authors · 2023-11-21
> **Response to Reviewer 5ZmP [key points: minimalistic code advances research objectives, adjust the observation accordingly]**
>
> Dear Reviewer 5ZmP,
>
> We sincerely thank you for your time and effort in reviewing our paper. We are encouraged by your comments that it 'solves a very important problem in the fairness literature', 'well states the challenges', and 'includes extensive experiments and summarizes their observations'. Most importantly, we value your suggestions and address them point by point in the following:
>
>
>
> **Q1: it would be better if the paper could provide more explanations on the strengths of FFB itself. It seems some explained characteristics (like minimalistic aspect) are not supported by enough convincing explanations**
>
> *---Since Q2 is based on the answer to this question, we will address it first.---*
>
> We thank you for these great and valuable comments. The term 'minimalistic' exemplifies our philosophy in this work: that code for research should be both simple and free from redundant elements. In this work, we firmly adhere to and follow this philosophy. This guiding principle also significantly shapes and differentiates FFB.
>
> - **Emphasis on Code Simplicity**: The FFB embodies our philosophy that simplicity is paramount in research code. Its design is deliberately straightforward, including the dataset loading function and metric functions, which can be easily understood without documentation. This simplicity not only facilitates modifications and extensions but also promotes seamless integration into one's own research.
>
> - **Facilitation of Research Objectives**: Adhering to these principles, FFB effectively furthers our goal of making research more accessible and focused. Researchers can concentrate on their primary objectives, avoiding complex or redundant code.
>
>
> Following the above two, we achieve our research goal that
> - **Comprehensive Evaluation of Commonly Used Fairness Methods**: We provide extensive experimental evidence to support previous observations, such as the instability of adversarial training and the utility-fairness trade-off. In addition, we also make new findings, including the presence of fairness in widely-used datasets and strategies for determining training stop.
>
>
>
> **Q2: The paper does not clearly explain how to use FFB for testing new algorithms or new datasets.**
>
> We thank you for your valuable suggestion. Our code follows our philosophy that simplicity is paramount in research code, making it easy to use FFB for testing new algorithms or datasets as follows:
>
> 1. **Testing New Algorithms:** The FFB is designed with a straightforward code structure, making it easy and accessible for integrating new algorithms. To test a new algorithm, you should follow the existing method style and use the dataset function and metric function.
>
> 2. **Incorporating New Datasets:** The FFB features a user-friendly Dataset loading API that offers flexibility to accommodate more datasets. To integrate new data, the function *load_XXXXX_data* serves as an initial step for incorporating new datasets. Below is the basic structure of the data loading function:
>      ```python
>      def load_XXXXX_data(path="../datasets", sensitive_attribute="sex"):
>          # Code to load and process new dataset
>          # X represents the features, y is the target variable, and s is the sensitive attribute
>          return X, y, s
>      ```
> 3. **Comprehensive Fairness Metrics:** The FFB includes a wide range of fairness metrics, which, to the best of our knowledge, are the most comprehensive. They provide a thorough assessment of algorithm performance across different dimensions of fairness. These metrics can be easily used to evaluate both existing and new algorithms on various datasets and adapted to other research projects.
>
>
>
>
> **Q3: the paper explains their observations on several algorithms and datasets by using FFB, but many of the observations are not very surprising and already discussed in the literature..**
>
> In our paper, we acknowledge that several observations made might not be entirely new. We have carefully reviewed and removed some observations that are well-established in the field. Instead, our focus has shifted to highlighting those findings that have not been explicitly detailed in previous works, as follows:
> - ➀ Not all widely used fairness datasets stably exhibit fairness issues.
> - ➁ Utility-fairness trade-offs are generally controllable by the hyperparameter.
> - ➂ The HSIC achieves the best utility-fairness trade-off among the tested methods.
> - ➃ Stopping training while the learning rate is lower enough is effective.
>
> We also modified our paper about the findings from our experiments on Page 2.
>
>
>
> We hope our response adequately addresses your concerns. If so, we kindly request the reviewer to re-evaluate our work and consider raising their score.
>
>
> Thanks,\
> Authors

---

> > ### Comment · Reviewer_5ZmP · 2023-11-23
> > **Thank you for the response**
> >
> > I appreciate the authors' response to my comments.
> > As some of my concerns are addressed, I have updated the score.
> >
> > However, I agree with other reviewers' comments that pointed out that the paper missed much state-of-the-art. For example, reweighting or sampling-based training methods are widely used in the fairness literature, as noted in other reviews. Thus, some messages may still be misleading.
> >
> > I understand it is not easy to add new baselines for all experiments; however, I would strongly recommend the authors at least add a better Related Work section 1) to cover more state-of-the-art fairness methods and 2) to clarify the scope of this paper's observations.

---

### Comment · Reviewer_dSax · 2023-11-19
**Stopping Review**

Hello everybody,

on Friday I had an emergency medical intervention on my eyes and I am now unable to read for the next few weeks. This message is also written with an assistance.

Overall, I cannot further contribute to the review process.

All the best

---

> ### Author Response · Authors · 2023-11-21
> **Hope you recover soon**
>
> Dear Reviewer dSax,
>
> I'm sorry to hear you had an emergency medical intervention on your eyes. I hope the intervention was successful and you have a full recovery.
>
> Take good care of yourself and I look forward to hearing from you whenever you have fully recovered.
>
> Sincerely,\
> Authors

---

### Author Response · Authors · 2023-11-22
**Summary of Revisions**

Dear Reviewers,

We sincerely appreciate the time and effort you've devoted to reviewing our work. We understand that your schedule may be quite busy. As the authors-reviewer discussion phase draws to a close, we kindly request your attention to our responses. Our aim is to gain insights into whether our responses effectively address your concerns and to ascertain if there are any additional questions or points you would like to discuss.

Based on your suggestions and feedback, we have taken the following steps to address your comments.

- We have revised our observations to remove the previously found observation and to highlight our new finding on Page 2.
- We clarify the scope of our current version, mainly focusing on the selection of baseline methods:
    - **Benchmarking Most Commonly Used Methods:** Our decision to concentrate on the most commonly used methods and employing these well-accepted baseline methods provides a reliable benchmark.
    - **In-Depth Analysis:** Our experimental design was not merely a surface-level examination of numerous methods. We conducted extensive experiments with these chosen methods to ensure a comprehensive and nuanced understanding.
- We restate the main characteristics of our FFB that differentiate it:
    - **Emphasis on Code Simplicity**: The FFB embodies our philosophy that simplicity is paramount in research code. Its design is deliberately straightforward, including the dataset loading function and metric functions, which can be easily understood without documentation.
    - **Facilitation of Research Objectives**: Adhering to these principles, FFB effectively furthers our goal of making research more accessible and focused. Researchers can concentrate on their primary objectives, avoiding complex or redundant code.
    - **Comprehensive Evaluation of Commonly Used Fairness Methods**: We provide extensive experimental evidence to support previous observations, such as the instability of adversarial training and the utility-fairness trade-off. In addition, we also make new findings, including the presence of fairness in widely-used datasets and strategies for determining training stopping.

We hope our response addresses your concerns. We look forward to the opportunity for further discussion with you. Thank you for your thoughtful consideration.

Thanks,\
Authors

---

### Meta-Review · Area_Chair_CsMZ · 2023-12-12

**Metareview:**

This paper introduces an open source benchmark for in-processing fair ML -- one of the three major approaches to ensuring model-specific quantitative fairness in ML (pre-, in-, and post-).  Reviewers appreciated the motivation, execution, reproducibility, and (to some extent) comprehensiveness of the benchmark.  Yet, as mentioned by reviewers in the discussion phase, I'd encourage the authors to continue expanding (i) the related work section to truly cover the space and (ii) to any extent possible, build out the set of benchmarks to completely cover the space.  For (i), as a review paper cross benchmark paper, this is especially important.  For (ii), I realize it's impossible to completely blanket the space of possible benchmarks, but the community would benefit from your fleshing this out to the extent possible prior to publication, and making it as easy as possible for the community to help you keep this codebase up to date.  Toward that end, I applaud your open sourcing everything, and encourage you to spend substantial time making it as easy as possible for the community to contribute to this codebase/benchmark.

**Justification For Why Not Higher Score:**

It's a solid paper - good motivation, good codebase, good writing.  It's unclear what a spotlight/oral would add to the work.

**Justification For Why Not Lower Score:**

Reviewer support, motivation.

---

### Decision · Program_Chairs · 2024-01-16

Accept (poster)